# Differentiable Annealed Importance Sampling and the Perils of Gradient Noise

**Guodong Zhang**[1,2]**, Kyle Hsu**[3]**, Jianing Li**[1]**, Chelsea Finn**[3]**, Roger Grosse**[1,2]
[1]University of Toronto, [2]Vector Institute, [3]Stanford University
{gdzhang, rgrosse}@cs.toronto.edu
{kylehsu, cbfinn}@cs.stanford.edu, jrobert.li@mail.utoronto.ca

## Abstract

Annealed importance sampling (AIS) and related algorithms are highly effective tools for marginal likelihood estimation, but are not fully differentiable due to the use of Metropolis-Hastings correction steps. Differentiability is a desirable property as it would admit the possibility of optimizing marginal likelihood as an objective using gradient-based methods. To this end, we propose Differentiable AIS (DAIS), a variant of AIS which ensures differentiability by abandoning the Metropolis-Hastings corrections. As a further advantage, DAIS allows for mini-batch gradients. We provide a detailed convergence analysis for Bayesian linear regression which goes beyond previous analyses by explicitly accounting for the sampler not having reached equilibrium. Using this analysis, we prove that DAIS is consistent in the full-batch setting and provide a sublinear convergence rate. Furthermore, motivated by the problem of learning from large-scale datasets, we study a stochastic variant of DAIS that uses mini-batch gradients. Surprisingly, stochastic DAIS can be arbitrarily bad due to a fundamental incompatibility between the goals of last-iterate convergence to the posterior and elimination of the accumulated stochastic error. This is in stark contrast with other settings such as gradient-based optimization and Langevin dynamics, where the effect of gradient noise can be washed out by taking smaller steps. This indicates that annealing-based marginal likelihood estimation with stochastic gradients may require new ideas.

## 1  Introduction

Marginal likelihood (ML), sometimes called *evidence*, is a central quantity in Bayesian learning as it measures how well a model can describe a particular dataset. It is commonly used to select hyperparameters for Gaussian processes [Rasmussen, 2003], where either closed-form solutions or accurate, tractable approximations are available. However, it is more often the case that computing ML is computationally intractable, as it involves summation or integration over high-dimensional model parameters or latent variables. In this case, one must resort to numerical methods or other approximations [Kass and Raftery, 1995]. In the context of model comparison (e.g., evaluating generative models [Wu et al., 2016, Huang et al., 2020]), annealed importance sampling (AIS) [Neal, 2001] is one of the most popular and effective algorithms. Notably, AIS is closely related to other generic ML estimators that yield accurate estimation [Grosse et al., 2015], including Sequential Monte Carlo (SMC) [Doucet et al., 2001] and nested sampling [Skilling et al., 2006]. Under some assumptions, AIS is able to produce accurate estimates of marginal likelihood given enough computation time (it converges to the true ML value quickly by adding more intermediate distributions).

AIS alternates between Markov chain Monte Carlo (MCMC) transitions and importance sampling updates, where the MCMC step typically involves a non-differentiable Metropolis-Hastings (MH) correction. Unfortunately, the non-differentiability precludes gradient-based optimization of the sampler and complicates theoretical analysis. To deal with this, we marry AIS with Hamiltonian

35th Conference on Neural Information Processing Systems (NeurIPS 2021).

Monte Carlo (HMC) [Neal, 2011] and derive an unbiased yet differentiable ML estimator named differentiable AIS (DAIS) by removing the MH correction step, which further unlocks the possibility of mini-batch computation. Moreover, DAIS can be made memory efficient by caching noise and simulating Hamiltonian dynamics in reverse [Maclaurin et al., 2015]. We analyze the convergence of DAIS in the setting of Bayesian linear regression. Our analysis goes beyond prior analyses of AIS in that we account for the sampler not having reached equilibrium. In the full-batch setting, we show that DAIS retains the consistency guarantee of AIS (despite the lack of MH steps) and has a sublinear convergence rate.

Furthermore, motivated by the problem of learning from large-scale datasets, we study a stochastic variant of our algorithm that uses gradients estimated from a subset of the dataset. Given the success of stochastic optimization [Robbins and Monro, 1951, Bottou and Bousquet, 2011] and stochastic gradient MCMC algorithms [Welling and Teh, 2011, Chen et al., 2014], one may presume that stochastic gradient DAIS would perform well. Surprisingly, the stochastic version of DAIS can be arbitrarily bad. In particular, we show that the log ML estimates of DAIS with stochastic gradients are inconsistent due to a fundamental incompatibility between the goals of last-iterate convergence to the posterior and elimination of the accumulated stochastic error. This is in stark contrast with other settings such as gradient-based optimization and Langevin dynamics, where the gradient noise can be washed out by taking smaller steps. This indicates that annealing-based ML estimation with minibatch gradients may require new ideas.

We validate our theoretical analysis with simulations. We also demonstrate empirically that DAIS can be applied to variational autoencoders (VAEs) [Kingma and Welling, 2013, Rezende et al., 2014] for a tighter evidence lower bound, which in turn leads to improved performance compared to the vanilla VAE. We also compare to importance weighted autoencoders (IWAE) [Burda et al., 2016]. While IWAE is more effective with a low compute budget, we show that DAIS eventually outperforms IWAE as compute increases. Finally, like AIS, DAIS can be used to evaluate generative models. We show that it performs on par with AIS despite the removal of the MH correction step and outperforms the IWAE bound by a large margin.

## 2    Background

### 2.1    Marginal Likelihood Estimation

For a model $\mathcal{M}$ and observed data $\mathcal{D}$, one can define the marginal likelihood (ML) as

$$p(\mathcal{D}|\mathcal{M}) = \int p(\mathcal{D}, \boldsymbol{\theta}|\mathcal{M})d\boldsymbol{\theta} = \int p(\mathcal{D}|\boldsymbol{\theta}, \mathcal{M})p(\boldsymbol{\theta}|\mathcal{M})d\boldsymbol{\theta}, \tag{1}$$

where $\boldsymbol{\theta}$ denotes the parameters of the model. ML estimation can be regarded as an instance of estimating the partition function of an unnormalized distribution. Given a distribution defined as $p(\boldsymbol{\theta}) = f(\boldsymbol{\theta})/\mathcal{Z}$ where the unnormalized density $f(\boldsymbol{\theta})$ can be efficiently computed, we are interested in estimating the partition function $\mathcal{Z} = \int f(\boldsymbol{\theta})d\boldsymbol{\theta}$. Here, $f(\boldsymbol{\theta})$ corresponds to $p(\mathcal{D}, \boldsymbol{\theta}|\mathcal{M})$ in (1). In this paper, we focus on the setting of ML estimation because the stochastic version of DAIS is most naturally viewed as arising from data subsampling, but otherwise our analysis applies to the more general setting.

It is often the case that computing ML is computationally intractable. One approach is to approximate (1) with Monte Carlo methods, e.g., one can approximate the integration using importance sampling:

$$p(\mathcal{D}|\mathcal{M}) = \mathbb{E}_{q(\boldsymbol{\theta})}\left[\frac{p(\mathcal{D}|\boldsymbol{\theta}, \mathcal{M})p(\boldsymbol{\theta}|\mathcal{M})}{q(\boldsymbol{\theta})}\right] \approx \frac{1}{S}\sum_{i=1}^{S}\frac{p(\mathcal{D}|\boldsymbol{\theta}_i, \mathcal{M})p(\boldsymbol{\theta}_i|\mathcal{M})}{q(\boldsymbol{\theta}_i)} \quad \text{with } \boldsymbol{\theta}_i \sim q(\boldsymbol{\theta}) \tag{2}$$

However, this estimation can exhibit high variance for small or medium $S$ when the target distribution $p(\mathcal{D}, \boldsymbol{\theta}|\mathcal{M})$ and proposal distribution $q(\boldsymbol{\theta})$ are dissimilar.

### 2.2    Annealed Importance Sampling

Annealed importance sampling (AIS) is an algorithm which estimates the ML by gradually changing, or "annealing", a distribution. Formally, the algorithm takes in a sequence of distributions $p_0, \ldots, p_K$, with $p_k(\boldsymbol{\theta}) = f_k(\boldsymbol{\theta})/\mathcal{Z}_k$ and $\mathcal{Z}_k = \int f_k(\boldsymbol{\theta})d\boldsymbol{\theta}$. In the context of ML estimation, the starting

distribution $f_0$ is the tractable prior distribution $p(\boldsymbol{\theta}|\mathcal{M})$ with $\mathcal{Z}_0 = 1$, while the target distribution $f_K$ is $p(\mathcal{D}, \boldsymbol{\theta}|\mathcal{M})$ with $\mathcal{Z}_K = p(\mathcal{D}|\mathcal{M})$. For each $p_k$, one must also specify an MCMC transition operator $\mathcal{T}_k$ which leaves $p_k$ invariant.

The output of AIS is an unbiased estimate $\hat{\mathcal{Z}}_K$ of the exact ML $\mathcal{Z}_K$. Importantly, unbiasedness holds for any finite $K$, as shown in Neal [2001]. Moreover, AIS can be viewed as importance sampling over an extended space [Neal, 2001]. In particular, we have $\mathcal{Z}_K = \mathbb{E}_{q_{\text{fwd}}}[q_{\text{bwd}}/q_{\text{fwd}}]$ with the target and proposal distributions defined as

$$q_{\text{fwd}}(\boldsymbol{\theta}_{0:K}) = p_0(\boldsymbol{\theta}_0)\mathcal{T}_1(\boldsymbol{\theta}_1|\boldsymbol{\theta}_0)\cdots\mathcal{T}_K(\boldsymbol{\theta}_K|\boldsymbol{\theta}_{K-1}) \tag{3}$$

$$q_{\text{bwd}}(\boldsymbol{\theta}_{0:K}) = f_K(\boldsymbol{\theta}_K)\tilde{\mathcal{T}}_K(\boldsymbol{\theta}_{K-1}|\boldsymbol{\theta}_K)\cdots\tilde{\mathcal{T}}_1(\boldsymbol{\theta}_0|\boldsymbol{\theta}_1), \tag{4}$$

where $\mathcal{T}_k(\boldsymbol{\theta}|\boldsymbol{\theta}')$ is a forward MCMC kernel and $\tilde{\mathcal{T}}_k(\boldsymbol{\theta}'|\boldsymbol{\theta}) = \mathcal{T}_k(\boldsymbol{\theta}|\boldsymbol{\theta}')p_k(\boldsymbol{\theta}')/p_k(\boldsymbol{\theta})$ is the corresponding reverse kernel. Here, $q_{\text{fwd}}$ represents the chain of states generated by AIS, and $q_{\text{bwd}}$ is a fictitious (unnormalized) reverse chain which begins with a sample from $p_K$ and applies the transitions in reverse order. In practice, the intermediate distributions have to be chosen carefully for a low variance estimate $\hat{\mathcal{Z}}_K$. One typically uses geometric averages of the initial and target distributions:

$$p_k(\boldsymbol{\theta}) = p_{\beta_k}(\boldsymbol{\theta}) = f_{\beta_k}(\boldsymbol{\theta})/\mathcal{Z}_{\beta_k} = f_0(\boldsymbol{\theta})^{1-\beta_k}f_K(\boldsymbol{\theta})^{\beta_k}/\mathcal{Z}_{\beta_k} = p(\boldsymbol{\theta}|\mathcal{M})p(\mathcal{D}|\boldsymbol{\theta}, \mathcal{M})^{\beta_k}/\mathcal{Z}_{\beta_k} \tag{5}$$

where $0 = \beta_0 < \beta_1 < \cdots < \beta_K = 1$ is the annealing schedule. Indeed, AIS gives an unbiased estimate $\hat{\mathcal{Z}}$ of $\mathcal{Z}$. However, as $\mathcal{Z}$ can vary over many orders of magnitude, it is often more meaningful to talk about estimating $\log \mathcal{Z}$. Unfortunately, unbiased estimators of $\mathcal{Z}$ can result in biased estimators of $\log \mathcal{Z}$ because $\mathbb{E}\log\hat{\mathcal{Z}} \leq \log\mathbb{E}\hat{\mathcal{Z}}$ by Jensen's inequality, resulting in only a lower bound. In particular, we have the AIS bound

$$\mathbb{E}_{q_{\text{fwd}}} \log \hat{\mathcal{Z}}_K = \sum_{k=1}^{K} \mathbb{E}_{q_{\text{fwd}}}\left[\log f_{\beta_k}(\boldsymbol{\theta}_{k-1}) - \log f_{\beta_{k-1}}(\boldsymbol{\theta}_{k-1})\right] \tag{6}$$

$$= \sum_{k=1}^{K}(\beta_k - \beta_{k-1})\mathbb{E}_{q_{\text{fwd}}}\left[\log p(\mathcal{D}|\boldsymbol{\theta}_{k-1}, \mathcal{M})\right] \tag{7}$$

where (5) facilitated the simplification from (6) to (7). Of course, it is not enough to have a lower bound; we would also like the estimates to be close to the true value. Fortunately, AIS is *consistent* in that the estimate $\log \hat{\mathcal{Z}}$ converges to the correct value in the limit of infinitely many intermediate distributions [Neal, 2001], under the very idealized assumption of *perfect transitions* (i.e. that each transition $\mathcal{T}_k$ generates an exact sample from $p_k$, independent of the previous state). To give some intuition, the bound (6) can be simplified as

$$\mathbb{E}_{q_{\text{fwd}}} \log \hat{\mathcal{Z}}_K = \log \mathcal{Z}_K - \sum_{k=1}^{K} D_{\text{KL}}(p_{k-1} \,\|\, p_k), \tag{8}$$

and the sum of the KL divergence terms diminishes as $K \to \infty$.

## 3 Differentiable Annealed Importance Sampling

In this section, we motivate and derive a differentiable AIS (DAIS) algorithm for marginal likelihood (ML) estimation. We also discuss its application to variational inference for a tighter ELBO and a memory-efficient implementation.

Ideally, assuming a continuously parameterized model class (e.g., variational autoencoder), we would like to differentiate through the lower bound (7) to find an optimal model $\mathcal{M}$. However, AIS must be instantiated with an MCMC transition kernel $\mathcal{T}_k$ that satisfies detailed balance to ensure it leaves $p_k$ invariant. In practice, this is typically achieved by using a MH step,

---

**Algorithm 1** Differentiable AIS (DAIS)

$\boldsymbol{\theta}_0, \mathbf{v}_0$ sample from $p_0(\boldsymbol{\theta}), \pi \triangleq \mathcal{N}(\mathbf{0}, \mathbf{M})$
$\mathcal{L}_{\text{DAIS}} = -\log p_0(\boldsymbol{\theta}_0)$
**for** $k = 1, \ldots, K$ **do**
$\quad \boldsymbol{\theta}_{k-\frac{1}{2}} \leftarrow \boldsymbol{\theta}_{k-1} + \frac{\eta}{2}\mathbf{M}^{-1}\mathbf{v}_{k-1}$
$\quad \hat{\mathbf{v}}_k \leftarrow \mathbf{v}_{k-1} + \eta\nabla\log f_{\beta_k}(\boldsymbol{\theta}_{k-\frac{1}{2}})$
$\quad \boldsymbol{\theta}_k \leftarrow \boldsymbol{\theta}_{k-\frac{1}{2}} + \frac{\eta}{2}\mathbf{M}^{-1}\hat{\mathbf{v}}_k$
$\quad \mathbf{v}_k \leftarrow \gamma\hat{\mathbf{v}}_k + \sqrt{1-\gamma^2}\boldsymbol{\varepsilon}, \ \boldsymbol{\varepsilon} \sim \mathcal{N}(\mathbf{0}, \mathbf{M})$
$\quad \mathcal{L}_{\text{DAIS}} \mathrel{+}= \log\pi(\hat{\mathbf{v}}_k)) - \log\pi(\mathbf{v}_{k-1})$
**end for**
**return** $\mathcal{L}_{\text{DAIS}} \mathrel{+}= \log p(\mathcal{D}, \boldsymbol{\theta}_K|\mathcal{M})$

---

which is generally not differentiable.[1] We thus remove the MH correction and, in particular, specify each transition to consist of a deterministic leapfrog integration step followed by a stochastic partial momentum refreshment [Horowitz, 1991]. Algorithm 1 details the simulation of Hamiltonian dynamics using such transitions. With $\gamma = 0$, the algorithm resembles unadjusted Langevin dynamics [Roberts et al., 1996] but computes the ML bound on the fly. In practice, choosing $0 < \gamma < 1$ ($\gamma = 0.9$ is a common default) helps avoid random walk behavior and accelerates mixing [Neal, 2011, Chen et al., 2014]. Importantly, we retain the formalism of performing importance sampling on an extended space despite the loss of detailed balance.[2] To show this, we can define the forward and (unnormalized) backward distributions as

$$q_{\text{fwd}}(\boldsymbol{\theta}_{0:K}, \mathbf{v}_{0:K}) = p_0(\boldsymbol{\theta}_0)\pi(\mathbf{v}_0)\mathcal{T}_1(\boldsymbol{\theta}_1, \mathbf{v}_1|\boldsymbol{\theta}_0, \mathbf{v}_0)\cdots\mathcal{T}_K(\boldsymbol{\theta}_K, \mathbf{v}_K|\boldsymbol{\theta}_{K-1}, \mathbf{v}_{K-1}) \tag{9}$$

$$q_{\text{bwd}}(\boldsymbol{\theta}_{0:K}, \mathbf{v}_{0:K}) = f_K(\boldsymbol{\theta}_K)\pi(\mathbf{v}_K)\tilde{\mathcal{T}}_K(\boldsymbol{\theta}_{K-1}, \mathbf{v}_{K-1}|\boldsymbol{\theta}_K, \mathbf{v}_K)\cdots\tilde{\mathcal{T}}_1(\boldsymbol{\theta}_0, \mathbf{v}_0|\boldsymbol{\theta}_1, \mathbf{v}_1) \tag{10}$$

where the transition operator $\mathcal{T}_k(\boldsymbol{\theta}_k, \mathbf{v}_k|\boldsymbol{\theta}_{k-1}, \mathbf{v}_{k-1}) = \mathcal{T}_k'(\boldsymbol{\theta}_k, \hat{\mathbf{v}}_k|\boldsymbol{\theta}_{k-1}, \mathbf{v}_{k-1})\mathcal{T}_k''(\mathbf{v}_k|\hat{\mathbf{v}}_k)$ is the composition of a leapfrog step and momentum refreshment step. We define the reverse chain by starting with an exact sample and executing each of the above steps of Algorithm 1 in the reverse order, which leads to a surprisingly simple expression for our estimator. In particular, the backward transition operator is defined by $\tilde{\mathcal{T}}_k(\boldsymbol{\theta}_{k-1}, \mathbf{v}_{k-1}|\boldsymbol{\theta}_k, \mathbf{v}_k) = \mathcal{T}_k''(\hat{\mathbf{v}}_k|\mathbf{v}_k)\mathcal{T}_k'(\boldsymbol{\theta}_{k-1}, \mathbf{v}_{k-1}|\boldsymbol{\theta}_k, -\hat{\mathbf{v}}_k)$. Note that we need to flip the sign of $\hat{\mathbf{v}}_k$ in the reverse chain to account for time reversal. As a consequence of the above definitions, we have

$$\mathcal{T}_k''(\mathbf{v}_k|\hat{\mathbf{v}}_k) = \mathcal{T}_k''(\hat{\mathbf{v}}_k|\mathbf{v}_k)\pi(\mathbf{v}_k)/\pi(\hat{\mathbf{v}}_k). \tag{11}$$

This is because $\mathcal{T}_k''(\mathbf{v}_k|\hat{\mathbf{v}}_k) = \mathcal{N}(\gamma\hat{\mathbf{v}}_k, (1-\gamma^2)\mathbf{M})$ and $\mathcal{T}_k''(\hat{\mathbf{v}}_k|\mathbf{v}_k) = \mathcal{N}(\gamma\mathbf{v}_k, (1-\gamma^2)\mathbf{M})$. Furthermore, since $\mathcal{T}_k'$ is a deterministic leapfrog update, it is reversible and volume preserving, so we have $\mathcal{T}_k'(\boldsymbol{\theta}_{k-1}, \mathbf{v}_{k-1}|\boldsymbol{\theta}_k, -\hat{\mathbf{v}}_k) = \mathcal{T}_k'(\boldsymbol{\theta}_k, \hat{\mathbf{v}}_k|\boldsymbol{\theta}_{k-1}, \mathbf{v}_{k-1})$. With this, we can derive the DAIS bound:

$$\mathcal{L}_{\text{DAIS}} = \mathbb{E}_{q_{\text{fwd}}}\left[\log q_{\text{bwd}}(\boldsymbol{\theta}_{0:K}, \mathbf{v}_{0:K}) - \log q_{\text{fwd}}(\boldsymbol{\theta}_{0:K}, \mathbf{v}_{0:K})\right]$$

$$= \mathbb{E}_{q_{\text{fwd}}}\left[\log p(\mathcal{D}, \boldsymbol{\theta}_K|\mathcal{M}) - \log p_0(\boldsymbol{\theta}_0) + \sum_{k=1}^{K}\log\frac{\pi(\hat{\mathbf{v}}_k)}{\pi(\mathbf{v}_{k-1})}\right]. \tag{12}$$

We remark that this bound supports the computation of pathwise derivatives.

It's useful to consider an intuition for the final term of (12), since this will help to clarify our convergence analysis. Observe that $\log\pi(\mathbf{v}_k)$ is simply the negative kinetic energy plus a constant, so this term will be negative if the kinetic energy increases over the course of a leapfrog step and positive if it decreases. For small enough step sizes, leapfrog steps approximately conserve the total energy. If the posterior distribution is becoming more concentrated over the course of annealing (as is typically the case), the start of the leapfrog step is likely to have atypically high potential energy, so the leapfrog step will convert the potential energy to kinetic energy, and this term will be negative in expectation. Conversely, if the distribution is becoming more spread out, kinetic energy will be converted to potential energy, and the term will be positive in expectation. Hence, when summed over the whole trajectory, this term helps to estimate the volume of the support of the posterior distribution.

## 3.1 Differentiable Annealed Variational Inference

DAIS can be applied to variational inference for a tighter bound; we name this differentiable annealed variational inference (DAVI). We note that the general idea of incorporating auxiliary MCMC states into a variational approximation was discussed in Salimans et al. [2015], but their formulation requires the specification and learning of a reverse transition model, whereas ours does not.

Recall that we can lower bound the log ML by choosing a tractable variational distribution and optimizing the bound. This has been widely adopted in variational autoencoders [Kingma and Welling, 2013, Rezende et al., 2014] and Bayesian neural networks [Blundell et al., 2015, Zhang et al., 2018]. The lower bound has the following form:

$$\mathcal{L} \equiv \mathbb{E}_{q_\phi}\left[\log p(\mathcal{D}, \boldsymbol{\theta}|\mathcal{M}) - \log q_\phi(\boldsymbol{\theta})\right] \tag{13}$$

---

[1]The discontinuity introduced by the MH step makes it hard to use the reparameterization trick, though this can be done using a delicate gradient estimator [Naesseth et al., 2017]. This is orthogonal to our work and our fix is simpler and easier for us to analyze.

[2]This is true even with mini-batch gradients.

However, the lower bound can be quite loose if the variational posterior family $q_\phi(\boldsymbol{\theta})$ is restrictive, e.g. Gaussian. To improve the bound, we can define a new variational distribution on an extended space as in (9), but starting from $q_\phi$ rather than $p_0$:

$$q_{\text{fwd}}(\boldsymbol{\theta}_{0:K}, \mathbf{v}_{0:K}) = q_\phi(\boldsymbol{\theta}_0)\pi(\mathbf{v}_0)\mathcal{T}_1(\boldsymbol{\theta}_1, \mathbf{v}_1|\boldsymbol{\theta}_0, \mathbf{v}_0)\cdots\mathcal{T}_K(\boldsymbol{\theta}_K, \mathbf{v}_K|\boldsymbol{\theta}_{K-1}, \mathbf{v}_{K-1}). \quad (14)$$

We also define associated intermediate distributions $p_k(\boldsymbol{\theta}) = q_\phi(\boldsymbol{\theta})^{1-\beta_k}p(\mathcal{D}, \boldsymbol{\theta}|\mathcal{M})^{\beta_k}$. This gives a new lower bound:

$$\mathcal{L}_{\text{DAVI}} \equiv \mathbb{E}_{q_{\text{fwd}}}\left[\log p(\mathcal{D}, \boldsymbol{\theta}_K|\mathcal{M}) - \log q_\phi(\boldsymbol{\theta}_0) + \sum_{k=1}^{K}\log\frac{\pi(\hat{\mathbf{v}}_k)}{\pi(\mathbf{v}_{k-1})}\right]. \quad (15)$$

We can maximize this lower bound over model parameters of $\mathcal{M}$, all parameters of AIS (e.g., annealing schedule $\beta_k$) as well as variational parameters $\phi$.

## 3.2 Memory-Efficient Implementation

Naively optimizing instantiations of (12) or (15) w.r.t. parameters using reverse-mode differentiation involves storing the entire sequence of sampled states $\boldsymbol{\theta}_0, \mathbf{v}_0, \ldots, \boldsymbol{\theta}_K, \mathbf{v}_K$. This can be problematic in cases when $K$ is large due to the large memory overhead. However, DAIS is compatible with the idea of reversible learning [Maclaurin et al., 2015], which ame-

**Table 1:** Memory and time usage of DAIS implementations. $B$ is 32 for single-precision floating-point format.

| Scheme | Memory | Time |
|---|---|---|
| Naive | $\mathcal{O}(BK)$ | $\mathcal{O}(K)$ |
| Rev. Learning | $\mathcal{O}(\log_2(1/\gamma)K)$ | $\mathcal{O}(K)$ |

liorates this problem. Instead of storing the states in memory, we can compute the previous state given the current state by reversing the dynamics. Recall that each DAIS transition is deterministic and reversible other than the use of noise $\boldsymbol{\varepsilon}_k$ for momentum refreshment. The exact noise samples can also be computed in reverse if one uses a deterministic and reversible scheme (e.g. the linear congruential generator) for managing pseudorandom number generator seeds. Assuming exact arithmetic (in practice, this is impossible), this means that the memory footprint of DAIS can be made *constant* with respect to the number of intermediate distributions $K$. Similar memory-efficiency tricks have also been used in other applications [Li et al., 2020, Ruan et al., 2021].

However, as discussed by Maclaurin et al. [2015], reversible learning with finite arithmetic precision requires some storage to counteract compounding round-off error. For $\gamma \neq 0$ ($\gamma = 0.9$ is a common default), we need on average $\log_2(1/\gamma)$ bits per parameter per step, which is still small compared to naive storage. We defer further exposition on memory-efficient DAIS to Appendix D. We remark that reversible learning is a potentially crucial property of DAIS as it affords some degree of scaling to longer chain lengths and, indirectly, bigger models.

## 4 Convergence Analysis for Bayesian Linear Regression

Neal [2001] has pointed that AIS is consistent, i.e. that it converges to the true log ML value in the limit of infinitely many intermediate distributions. However, these consistency results depend on the idealized assumption of perfect transitions (where each transition returns an independent exact sample), and therefore don't account for the time required for the samples to reach equilibrium. Here, we analyze DAIS for a Bayesian linear regression model with realistic (imperfect) transitions. Accounting for the convergence of the sampler is essential for separating the behaviors in the full-batch and mini-batch regimes.

In particular, we focus on the Bayesian linear regression setting and adopt the following model:

prior: $\boldsymbol{\theta} \sim \mathcal{N}(\boldsymbol{\mu}_p, \boldsymbol{\Lambda}_p^{-1})$

likelihood: $\mathbf{y} \sim \mathcal{N}(\mathbf{X}\boldsymbol{\theta}, \sigma^2\mathbf{I}) \Rightarrow \boldsymbol{\theta} \sim \mathcal{N}(\boldsymbol{\mu}_*, \boldsymbol{\Lambda}_{\text{lld}}^{-1})$ where $\boldsymbol{\Lambda}_{\text{lld}} = \frac{\mathbf{X}^\top\mathbf{X}}{\sigma^2}$ and $\boldsymbol{\mu}_* = (\mathbf{X}^\top\mathbf{X})^{-1}\mathbf{X}^\top\mathbf{y}$

posterior: $\boldsymbol{\theta} \sim \mathcal{N}(\boldsymbol{\mu}_{\text{pos}}, \boldsymbol{\Lambda}_{\text{pos}}^{-1})$ where $\boldsymbol{\mu}_{\text{pos}} = \boldsymbol{\Lambda}_{\text{pos}}^{-1}(\boldsymbol{\Lambda}_p\boldsymbol{\mu}_p + \boldsymbol{\Lambda}_{\text{lld}}\boldsymbol{\mu}_*)$ and $\boldsymbol{\Lambda}_{\text{pos}} = \boldsymbol{\Lambda}_p + \boldsymbol{\Lambda}_{\text{lld}}$

with $\mathbf{X} \in \mathbb{R}^{n \times d}$ denoting the input features and $\mathbf{y} \in \mathbb{R}^{n \times 1}$ the targets. We choose Bayesian linear regression because it enables us to analyze the dynamics analytically in a similar manner as done by the noisy quadratic model (NQM) [Zhang et al., 2019] in the context of optimization. We adopt the leapfrog step (we assume an identity mass matrix without loss of generality because we can absorb

$\mathbf{M}$ into the input matrix $\mathbf{X}$ in Algorithm 1) and obtain the following update rule (see Appendix B.1 for derivation):

$$
\boldsymbol{\theta}_k \leftarrow \left(\mathbf{I} - \frac{\eta_k^2}{2}\boldsymbol{\Lambda}_{\mathrm{pos}}^{\beta_k}\right)\boldsymbol{\theta}_{k-1} + \left(\eta_k\mathbf{I} - \frac{\eta_k^3}{4}\boldsymbol{\Lambda}_{\mathrm{pos}}^{\beta_k}\right)\mathbf{v}_{k-1} + \frac{\eta_k^2}{2}\boldsymbol{\Lambda}_{\mathrm{pos}}^{\beta_k}\boldsymbol{\mu}_{\mathrm{pos}}^{\beta_k}
$$
$$
\hat{\mathbf{v}}_k \leftarrow -\eta_k\boldsymbol{\Lambda}_{\mathrm{pos}}^{\beta_k}\boldsymbol{\theta}_{k-1} + \left(\mathbf{I} - \frac{\eta_k^2}{2}\boldsymbol{\Lambda}_{\mathrm{pos}}^{\beta_k}\right)\mathbf{v}_{k-1} + \eta_k\boldsymbol{\Lambda}_{\mathrm{pos}}^{\beta_k}\boldsymbol{\mu}_{\mathrm{pos}}^{\beta_k}
\tag{16}
$$

where $\boldsymbol{\Lambda}_{\mathrm{pos}}^{\beta_k} = \boldsymbol{\Lambda}_p + \beta_k\boldsymbol{\Lambda}_{\mathrm{lld}}$ and $\boldsymbol{\mu}_{\mathrm{pos}}^{\beta_k} = (\boldsymbol{\Lambda}_{\mathrm{pos}}^{\beta_k})^{-1}(\boldsymbol{\Lambda}_p\boldsymbol{\mu}_p + \beta_k\boldsymbol{\Lambda}_{\mathrm{lld}}\boldsymbol{\mu}_*)$. With these iterative updates, we can compute the expectation and covariance of $\boldsymbol{\theta}_k$ and $\mathbf{v}_k$ at any time $k$, which suffices to compute the lower bound in closed-form.

## 4.1 Sublinear Convergence in the Full-Batch Setting

With the model defined, we now show that our algorithm is asymptotically consistent, i.e., the bound (12) converges to exact log ML as $K$ goes to infinity. For Bayesian linear regression, the update rules in (16) are affine transformations of Gaussian random variables, so the distribution of $\boldsymbol{\theta}_k$ is also Gaussian in the form of $\mathcal{N}(\boldsymbol{\mu}_k, \boldsymbol{\Sigma}_k)$. We can compute the gap between the log ML and our lower bound in closed-form (see Appendix B.2 for derivation):

$$
\log p(\mathcal{D}) - \mathcal{L}_{\mathrm{DAIS}} =
$$
$$
\underbrace{\frac{1}{2}\|\boldsymbol{\mu}_K - \boldsymbol{\mu}_{\mathrm{pos}}\|^2_{\boldsymbol{\Lambda}_{\mathrm{pos}}}}_{\textcircled{1}} + \underbrace{\frac{1}{2}\mathrm{Tr}(\boldsymbol{\Lambda}_{\mathrm{pos}}\boldsymbol{\Sigma}_K) - \frac{d}{2}}_{\textcircled{2}} + \underbrace{\frac{1}{2}\log\frac{|\boldsymbol{\Sigma}_{\mathrm{pos}}|}{|\boldsymbol{\Sigma}_p|} - \mathbb{E}_q\left[\sum_{k=1}^K \log\frac{\pi(\hat{\mathbf{v}}_k)}{\pi(\mathbf{v}_{k-1})}\right]}_{\textcircled{3}}
\tag{17}
$$

where $d$ is the feature dimension. Here, $\textcircled{1}$ and $\textcircled{2}$ measure the error of last-iterate Markov chain convergence and will both vanish as long as $\boldsymbol{\mu}_K \to \boldsymbol{\mu}_{\mathrm{pos}}$ and $\boldsymbol{\Sigma}_K \to \boldsymbol{\Sigma}_{\mathrm{pos}}$. We will show later that they converge with a rate of $\mathcal{O}(\frac{1}{\eta^2 K})$. The key is to show that $\boldsymbol{\mu}_k$ (resp. $\boldsymbol{\Lambda}_k$) lags behind $\boldsymbol{\mu}_{\mathrm{pos}}^{\beta_k}$ (resp. $\boldsymbol{\Lambda}_{pos}^{\beta_k}$) with roughly $\frac{1}{\eta^2}$ steps. Formally, we have the following.

**Lemma 1.** *Given equally spaced $\beta_k$, running DAIS with $\gamma = 0$ and $\eta \sim \frac{1}{K^c}$ where $c \geq \frac{1}{4}$ yields*

$$
\|\boldsymbol{\mu}_{k-1} - \boldsymbol{\mu}_{pos}^{\beta_k}\|_2 = \mathcal{O}(K^{2c-1}), \quad \|\boldsymbol{\Lambda}_{k-1} - \boldsymbol{\Lambda}_{pos}^{\beta_k}\|_2 = \mathcal{O}(K^{2c-1}).
\tag{18}
$$

We remark that the assumption of $\beta_k$ being equally spaced is not essential and can be relaxed as long as they are chosen by a scheme that leads to $\beta_k - \beta_{k-1}$ going down approximately in inverse proportion to $K$. In addition, we note that the assumption of full momentum refreshment is for convenience and we believe a similar result holds for $\gamma > 0$.

Importantly, this lemma implies that both $\textcircled{1}$ and $\textcircled{2}$ vanish sublinearly if we choose $c < \frac{1}{2}$. The analysis of error term $\textcircled{3}$ is more nuanced. In particular, this error could either come from using transitions for each of these intermediate distributions that do not bring the distribution close to equilibrium, or from using a finite number of distributions to anneal from $p_0$ to $p_K$. Surprisingly, the error $\textcircled{3}$ decays as fast as the other two terms if the step size scales as $1/K^c$ with $c \geq \frac{1}{4}$. In summary, we have the following theorem.

**Theorem 1.** *Given equally spaced $\beta_k$, running DAIS with $\gamma = 0$ and $\eta \sim \frac{1}{K^c}$ where $c \geq \frac{1}{4}$ yields*

$$
\log p(\mathcal{D}) - \mathcal{L}_{DAIS} = \mathcal{O}(K^{2c-1}).
$$

*With $c = \frac{1}{4}$, we have the convergence rate $\mathcal{O}(1/\sqrt{K})$.*

We remark that with perfect transitions, the requirement of $c \geq 1/4$ is not necessary and we can achieve $\mathcal{O}(1/K)$ convergence, as also shown in Grosse et al. [2013]. The gap between $\mathcal{O}(1/\sqrt{K})$ and $\mathcal{O}(1/K)$ highlights the importance of considering convergence to the stationary distribution.

## 4.2 Inconsistency in the Stochastic Setting

Standard AIS involves MH steps which require a costly computation using all of the data, thus defeating the potential computational efficiency benefit of stochastic gradients [Chen et al., 2014].

This is likely why the convergence properties of stochastic gradient AIS were previously unknown. In contrast, the only part of DAIS that requires full-batch computation is the gradient term in the leapfrog step of Algorithm 1, which may be amenable to estimation via mini-batching.

We have shown that our algorithm is asymptotically consistent in the full-batch setting. Often, a consistent/convergent algorithm in the deterministic setting readily implies a similar convergence result in the stochastic setting. For example, SGD [Robbins and Monro, 1951] and SGMCMC [Chen et al., 2014, Ma et al., 2015] are both convergent in the presence of noise. This begs the question of whether DAIS is consistent when we only have access to stochastic gradients. Here, we adopt an additive noise model[3] $\tilde{\nabla} \log f_k(\boldsymbol{\theta}) = \nabla \log f_k(\boldsymbol{\theta}) + \boldsymbol{\varepsilon}$. This model is commonly used in the stochastic approximation literature, and such a model has also been adopted in Chen et al. [2014]. With such a noise model, we have the following dynamics:

$$
\begin{aligned}
\boldsymbol{\theta}_k &\leftarrow \left( \mathbf{I} - \frac{\eta_k^2}{2} \boldsymbol{\Lambda}_{\text{pos}}^{\beta_k} \right) \boldsymbol{\theta}_{k-1} + \left( \eta_k \mathbf{I} - \frac{\eta_k^3}{4} \boldsymbol{\Lambda}_{\text{pos}}^{\beta_k} \right) \mathbf{v}_{k-1} + \frac{\eta_k^2}{2} \boldsymbol{\Lambda}_{\text{pos}}^{\beta_k} \boldsymbol{\mu}_{\text{pos}}^{\beta_k} + \frac{\eta_k^2}{2} \boldsymbol{\varepsilon} \\
\hat{\mathbf{v}}_k &\leftarrow -\eta_k \boldsymbol{\Lambda}_{\text{pos}}^{\beta_k} \boldsymbol{\theta}_{k-1} + \left( \mathbf{I} - \frac{\eta_k^2}{2} \boldsymbol{\Lambda}_{\text{pos}}^{\beta_k} \right) \mathbf{v}_{k-1} + \eta_k \boldsymbol{\Lambda}_{\text{pos}}^{\beta_k} \boldsymbol{\mu}_{\text{pos}}^{\beta_t} + \eta_k \boldsymbol{\varepsilon}
\end{aligned}
\tag{19}
$$

where stochastic noise $\boldsymbol{\varepsilon}$ has variance lowered bounded by $\boldsymbol{\Sigma}_{\boldsymbol{\varepsilon}}$. Further, we let $\boldsymbol{\mu}_k^{\mathbf{v}} = \mathbb{E}[\hat{\mathbf{v}}_k]$ and $\boldsymbol{\Sigma}_k^{\mathbf{v}}$ be the covariance of $\hat{\mathbf{v}}_k$. Surprisingly, we find that DAIS is incompatible with stochastic gradients, even though it admits the same importance sampling interpretation if the reverse transition $\tilde{\mathcal{T}}_k$ conditions on the same mini-batch of data as $\mathcal{T}_k$. We summarize the result in the following theorem.

**Theorem 2.** *For stochastic DAIS with full momentum refreshment ($\gamma = 0$ in Algorithm 1) and any stepsize scheme, we have*

$$
\liminf_{K \to \infty} |\log p(\mathcal{D}) - \mathcal{L}_{DAIS}| > 0.
\tag{20}
$$

Here, we give some intuition why DAIS fails in the stochastic setting. To ensure convergence of $\boldsymbol{\theta}_K$ to $\mathcal{N}(\boldsymbol{\mu}_{\text{pos}}, \boldsymbol{\Sigma}_{\text{pos}})$, a major requirement is for the step sizes to satisfy $\lim_{K \to \infty} \sum_{k=1}^K \eta_k^2 = \infty$ [Robbins and Monro, 1951]. However, the randomness of mini-batching sampling would contribute to the variance of $\hat{\mathbf{v}}_k$. In particular, we have the following recursion:

$$
\tilde{\boldsymbol{\Sigma}}_k^{\mathbf{v}} = \eta_k^2 \boldsymbol{\Lambda}_{\text{pos}}^{\beta_k} \hat{\boldsymbol{\Sigma}}_{k-1} \boldsymbol{\Lambda}_{\text{pos}}^{\beta_k} + \left( \mathbf{I} - \frac{\eta_k^2}{2} \boldsymbol{\Lambda}_{\text{pos}}^{\beta_k} \right)^2 + \eta_k^2 \boldsymbol{\Sigma}_{\boldsymbol{\varepsilon}}.
\tag{21}
$$

For notational convenience, we let $\hat{\boldsymbol{\Sigma}}_k^{\mathbf{v}} \triangleq \eta_k^2 \boldsymbol{\Lambda}_{\text{pos}}^{\beta_k} \hat{\boldsymbol{\Sigma}}_{k-1} \boldsymbol{\Lambda}_{\text{pos}}^{\beta_k} + (\mathbf{I} - \frac{\eta_k^2}{2} \boldsymbol{\Lambda}_{\text{pos}}^{\beta_k})^2$. Here, we used $\tilde{\boldsymbol{\Sigma}}_k^{\mathbf{v}}$, $\hat{\boldsymbol{\Sigma}}_k^{\mathbf{v}}$ and $\hat{\boldsymbol{\Sigma}}_k$ to avoid confusion with $\boldsymbol{\Sigma}_k^{\mathbf{v}}$ and $\boldsymbol{\Sigma}_k$ in the full-batch setting. In this case, if we follow Stephan et al. [2017] and Chen et al. [2014] in assuming $\boldsymbol{\varepsilon}$ is Gaussian,[4] we have

$$
\mathbb{E}_q \left[ \sum_{k=1}^K \log \frac{\pi(\hat{\mathbf{v}}_k)}{\pi(\mathbf{v}_{k-1})} \right] = \sum_{k=1}^K \left[ -\frac{1}{2} \|\boldsymbol{\mu}_k^{\mathbf{v}}\|_2^2 - \frac{1}{2} \text{Tr}(\hat{\boldsymbol{\Sigma}}_k^{\mathbf{v}}) + \frac{d}{2} \right] - \sum_{k=1}^K \left[ \frac{1}{2} \eta_k^2 \text{Tr}(\boldsymbol{\Sigma}_{\boldsymbol{\varepsilon}}) \right].
\tag{22}
$$

The second term of (22) goes to infinity as $\lim_{K \to \infty} \sum_{k=1}^K \eta_k^2 = \infty$. Intuitively, the gradient noise adds to the kinetic energy, and the size of this contribution is proportional to $\eta_k^2$. Since this effect is cumulative over all $K$ steps, $\eta_k$ has to be reduced at least as $1/\sqrt{K}$ for the kinetic energy term to go down. However, this contradicts the requirement that $\lim_{K \to \infty} \sum_{k=1}^K \eta_k^2 = \infty$, needed for last-iterate convergence. In summary, the convergence of $\boldsymbol{\theta}_K$ requires us to make sure the sum of step sizes goes to infinity, which in turn results in a non-convergent kinetic energy term (22).

One may wonder why gradient noise does not hurt the convergence of SGLD [Welling and Teh, 2011] or SGMCMC [Ma et al., 2015]. Generally speaking, these algorithms are only concerned with the last iteration convergence to the true posterior, hence one can eliminate the stochastic error by taking more steps of a smaller size. In contrast, our bound (and potentially other AIS-style algorithms) relies on all intermediate distributions, and so the error induced by stochastic gradient noise accumulates over the whole trajectory. Simply taking smaller steps fails to reduce the error. We conjecture that AIS-style algorithms are inherently fragile to gradient noise.

---

[3] In reality, the noise consists of two parts: multiplicative input subsampling noise and additive label noise. We can assume that the noise is lower bounded by the additive part. See Appendix B.4 for justifications.

[4] This assumption is non-essential, though Gaussian noise is reasonable by the central limit theorem.

# 5   Related Works

For ML estimation (or partition function estimation), Sequential Monte Carlo (SMC) [Doucet et al., 2001, Del Moral et al., 2006] is another popular method which is derived from particle filtering. While SMC is based on a different intuition from AIS, the underlying mathematics is equivalent. In SMC, the intermediate distributions are defined by conditioning on a sequence of increasing subsets of data. Both AIS and SMC are closely related to a broader family of techniques for partition function estimation, all based on the following identity from statistical physics: $\log \mathcal{Z}_K - \log \mathcal{Z}_0 = \int_0^1 \mathbb{E}_{\boldsymbol{\theta} \sim p_\beta} \left[ \frac{d}{d\beta} \log f_\beta(\boldsymbol{\theta}) \right] d\beta$. In particular, the weight update in AIS can be seen as a finite difference approximation. In comparison, thermodynamic integration (TI) [Frenkel and Smit, 2001] estimates this integration using numerical quadrature, and path sampling [Gelman and Meng, 1998] does so with Monte Carlo integration. Recently, Masrani et al. [2019] connected TI and variational inference for a tighter bound on the log ML, but they computed each intermediate term using importance sampling rather than annealing-based sampling algorithms. More recently[5], Thin et al. [2021] proposed a similar algorithm based on sequential importance sampling with unadjusted Langevin kernels in the context of variational auto-encoders. Concurrently, Geffner and Domke [2021] proposed the same algorithm as our DAVI with a focus on the empirical side.

In the context of variational inference, many papers have also investigated tighter lower bounds for the log ML. Burda et al. [2016] proposed a strictly tighter log-likelihood lower bound derived from importance weighting and Luo et al. [2019] recently extended this idea of importance sampling to derive an unbiased (but potentially high variance) estimator of log ML using the Russian roulette estimator [Kahn, 1955]. Salimans et al. [2015] proposed to incorporate MCMC iterations into the variational approximation. The central idea is that we can interpret the stochastic Markov chain as part of a variational approximation in an extended space, as we did in the paper. However, the proposed methods require learning reverse kernels, which has a large impact on performance. The same authors also briefly discussed annealed variational inference, which combines variational inference and AIS. However, their derivation relies on the detailed balance assumption and is therefore not amenable to gradient-based optimization. Later, Caterini et al. [2019] proposed the Hamiltonian VAE, which improves Hamiltonian variational inference [Salimans et al., 2015] with an optimally chosen reverse MCMC kernel. In particular, they removed the momentum sampling step and used deterministic transitions. The resulting algorithm can be thought of as a normalizing flow scheme in which the flow depends explicitly on the target distribution. Along this line, Le et al. [2018], Naesseth et al. [2018], Maddison et al. [2017] proposed to meld variational inference and SMC for time-series models.

Finally, stochastic gradient variants of several MCMC algorithms [Welling and Teh, 2011, Chen et al., 2014, Ma et al., 2015] have been proposed over the last decade. In particular, these works showed that adding the "right amount" of noise to the parameter updates leads to samples from the target posterior as long as the step size is annealed. Importantly, the convergence rates of these algorithms are established in both the full-batch setting [Dalalyan, 2017, Cheng et al., 2018] and the stochastic setting [Chen et al., 2015, Teh et al., 2016, Raginsky et al., 2017, Zou et al., 2020]. By contrast, the convergence properties for AIS and related algorithms were largely unknown even for the deterministic case, and it remains largely unexplored whether AIS can be made compatible with stochastic gradients.

# 6   Simulations

In this section, we discuss the experiments used to validate our algorithm and theory. Importantly, we do *not* aim to achieve state-of-the-art on these tasks.

## 6.1   Bayesian Linear Regression

In Section 4, we proved for the Bayesian linear regression setting that while DAIS is asymptotically consistent with full-batch gradient, the noise injected into the system via stochastic gradients precludes convergence. Here, we verify our theory with numerical simulations. The $n$ input vectors $\mathbf{X} \in \mathbb{R}^{n \times d}$ and targets $\mathbf{y} \in \mathbb{R}^n$ respectively consist of entries sampled from $\mathcal{N}(0, 0.01)$ and $\mathcal{N}(0, 1)$. In particular, we choose $n = 10,000$ and $d = 10$ for our simulations (the results are qualitatively same

---

[5]The paper appeared on arXiv after we submitted the paper.

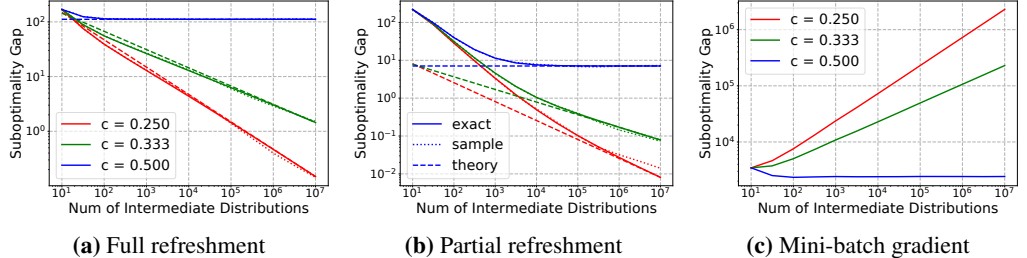

| | (a) Full refreshment | (b) Partial refreshment | (c) Mini-batch gradient |

**Figure 1:** Gap between true log ML and our DAIS bound as a function of number of intermediate distributions. Solid lines are exact computation of our DAIS bound; dotted lines are sample-based simulation (Monte Carlo method with 100 samples); dashed lines are theoretical predictions based on Theorem 1 with slope $2c - 1$. For the rightmost figure, we use a batch size of 100.

**Table 2:** Test negative log-likelihood of the trained model, estimated using AIS with 10,000 intermediate distribution and 10 particles. For VAE/IWAE, we used $S \times K$ samples. The numbers reported are averaged over three runs. The standard deviations are fairly small over three runs ($< 0.06$).

| Objective | $S \times K = 1$ | $S \times K = 5$ | $S \times K = 50$ | | | $S \times K = 500$ | |
|---|---|---|---|---|---|---|---|
| | $K = 1$ | $K = 5$ | $K = 5$ | $K = 10$ | $K = 50$ | $K = 10$ | $K = 50$ |
| VAE | 86.93 | 86.95 | | 86.94 | | | 86.89 |
| IWAE | 86.93 | **85.43** | | 84.46 | | | 83.87 |
| DAVI | - | 86.51 | 84.49 | 84.45 | 85.23 | 83.62 | 83.65 |
| DAVI (adapt) | - | 86.49 | 84.42 | **84.39** | 85.00 | **83.56** | 83.69 |

with different $n$ and $d$). In addition, we set the observation variance $\sigma^2 = 1$. For convenience, we set the linear annealing scheme $\beta_k = \frac{k}{K}$.

In Figure 1, we report the gap between exact log ML and our bound as a function of number of intermediate distributions. With full-batch gradients, the simulations (solid and dotted lines) align well with our theoretical predictions (dashed lines) for different step-size scaling schemes, suggesting our bound in Theorem 1 is tight. In addition, we observe in Figure 1c that the gap fails to vanish with mini-batch gradients for all step-size scaling schemes. Interestingly, with $c = 1/2$, the gap stays constant. This matches our predictions that the deterministic error decays as $\mathcal{O}(K^{2c-1}) = \mathcal{O}(1)$ while the stochastic error is proportional to $\sum_{k=1}^{K} \eta_k^2 = \mathcal{O}(K^{2c-1}) = \mathcal{O}(1)$.

### 6.2 Variational Autoencoder

We compare the performance of DAVI to vanilla VAE [Kingma and Welling, 2013] and IWAE [Burda et al., 2016] on density modeling tasks. We use the dynamically binarized MNIST [LeCun et al., 1998] dataset. We use the same architecture as in IWAE paper. The prior $p(\mathbf{z})$ is a 50-dimensional standard Gaussian distribution. The conditional distributions $p(\mathbf{x}_i|\mathbf{z})$ are independent Bernoulli, with the decoder parameterized by two hidden layers, each with 200 tanh units. The variational posterior $q(\mathbf{z}|\mathbf{x})$ is also a 50-dimensional Gaussian with diagonal covariance, whose mean and variance are both parameterized by two hidden layers with 200 tanh units (see other details in Appendix C.1).

In the first set of experiments, we investigate the effect of number of intermediate distributions $K$ and combine it with importance sampling (as done in IWAE) with $S$ samples in DAVI. To be specific, we define the bound as follows:

$$\log \frac{1}{S} \sum_{i=1}^{S} \left( \frac{p_{\boldsymbol{\theta}}(\mathbf{x}, \mathbf{z}_K^i)}{q_{\boldsymbol{\phi}}(\mathbf{z}_0^i|\mathbf{x})} \prod_{k=1}^{K} \frac{\pi(\hat{\mathbf{v}}_k^i)}{\pi(\mathbf{v}_{k-1}^i)} \right), \tag{23}$$

where we sample $(\mathbf{z}_0^i, \mathbf{v}_0^i, \hat{\mathbf{v}}_1^i, \dots)$ independently from $q_{\text{fwd}}$. By default, we use partial momentum refreshment with $\gamma = 0.9$ and equally spaced annealing parameters $\beta_k = k/K$. We compare it to vanilla VAE and IWAE bounds with $S \times K$ samples. As shown in Table 2, increasing $K$ gives strictly better models with lower test negative log-likelihood. However, IWAE achieves slightly better performance with roughly the same computation if $S \times K$ is small. On the other hand, DAVI is more effective with more compute budget (i.e., $S \times K$ is large) and eventually outperforms IWAE.

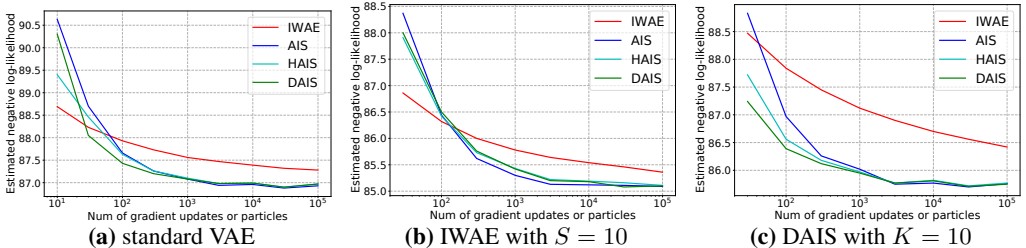

**Figure 2:** Results of different algorithms in evaluating VAE-, IWAE- or DAIS-trained models. DAIS performs on par with AIS/HAIS but without requiring the MH correction steps.

In the second set of experiments, we learn the annealing scheme of DAVI together with the parameters of encoder and decoder. Comparing the third and fourth rows of Table 2, one can see that learning the annealing scheme improves the performance slightly.

Lastly, we also compare our algorithm with IWAE, AIS, and Hamiltonian AIS (HAIS)[6] [Sohl-Dickstein and Culpepper, 2012] in evaluating the log-likelihood of trained models. Notably, IWAE and AIS have been widely used in VAE evaluation, see e.g. Wu et al. [2016], Huang et al. [2020]. For HAIS and DAIS, we employ the optimal step-size scaling scheme derived in Theorem 1 with $c = 1/4$ and only tune the step size for the case of $K = 10$. For all implementation details, please see Appendix C.2. In particular, we compare the evaluation algorithms on models trained using the VAE, IWAE, and DAIS objectives. In Figure 2, we report the estimated negative log-likelihood as a function of the number of particles (for IWAE) or gradient updates (for AIS, HAIS and DAIS). Interestingly, we observe that IWAE performs better when we have limited computation and AIS/HAIS/DAIS win out by a big margin if we increase $K$. Indeed, it is known that naive importance sampling can have exponential sample complexity in some problem parameters (e.g. the dimension), and that AIS can overcome this to efficiently give accurate results (see, for instance, the analysis in Neal [2001]). This may explain the superior performance of AIS-based algorithms when more compute is used. Importantly, we observe that DAIS performs on par with AIS/HAIS but without requiring the MH correction steps.

## 7 Conclusion

In this paper, we proposed a differentiable AIS (DAIS) algorithm for marginal likelihood estimation. We provided a detailed convergence analysis for Bayesian linear regression which goes beyond existing analyses. Using this, we proved a sublinear convergence rate of DAIS in the full-batch setting. However, we showed that DAIS is inconsistent when mini-batch gradients are used due to a fundamental incompatibility between the goals of last-iterate convergence to the posterior and elimination of the pathwise stochastic error. This comprises an interesting counterexample to the general trend of algorithms consistent in the deterministic setting remaining consistent in the stochastic setting. Our negative result helps explain the difficulty of developing practically effective AIS-like algorithms that exploit mini-batch gradients. Our numerical experiments validate our claims.

## Acknowledgements

We thank Ricky Tian Qi Chen, Murat A. Erdogdu, Sergey Levine, Xuechen Li, Mufan Li, Chris J. Maddison, and Matthew D. Hoffman for many helpful discussions. In particular, we thank Xuechen Li for his suggestion regarding a memory-efficient implementation. We also thank Shengyang Sun, Xuechen Li and Yangjun Ruan for detailed comments on early drafts, and Radford M. Neal for insightful feedback on our arXiv preprint. Finally, we thank the anonymous NeurIPS reviewers for their useful feedback on earlier versions of this manuscript.

GZ was supported by Ontario Graduate Fellowship. KH was supported by a Sequoia Capital Stanford Graduate Fellowship. RG was supported by the CIFAR AI Chairs program. Resources used in preparing this research were provided, in part, by the Province of Ontario, the Government of Canada through CIFAR, and companies sponsoring the Vector Institute.

---

[6]HAIS is an algorithm similar to ours which combines Hamiltonian dynamics with AIS, but with accept-reject steps included.

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
