*Proof.* We will prove the result for both quantities by induction. We assume $\boldsymbol{\mu}_p = \mathbf{0}$ and $\boldsymbol{\Lambda}_p \succeq \mathbf{I}$ without loss of generality and choose $\eta = \frac{a}{K^c}$. Recall definitions

$$\boldsymbol{\Lambda}_{\text{lld}} = \frac{1}{\sigma^2} \mathbf{X}^\top \mathbf{X} \tag{24}$$

$$\boldsymbol{\Lambda}_{\text{pos}}^{\beta_k} = \boldsymbol{\Lambda}_p + \beta_k \boldsymbol{\Lambda}_{\text{lld}} \tag{25}$$

$$\boldsymbol{\mu}_* = \left(\mathbf{X}^\top \mathbf{X}\right)^{-1} \mathbf{X}^\top \mathbf{y} \tag{26}$$

$$\boldsymbol{\mu}_{\text{pos}}^{\beta_k} = \left(\boldsymbol{\Lambda}_{\text{pos}}^{\beta_k}\right)^{-1} \left(\boldsymbol{\Lambda}_p \boldsymbol{\mu}_p + \beta_k \boldsymbol{\Lambda}_{\text{lld}} \boldsymbol{\mu}_*\right). \tag{27}$$

We first bound intermediate quantities of interest. In particular, we have

$$\begin{aligned}
\|\boldsymbol{\Sigma}_{\text{pos}}^{\beta_{k-1}} - \boldsymbol{\Sigma}_{\text{pos}}^{\beta_k}\|_2 &\leq \|\boldsymbol{\Sigma}_{\text{pos}}^{\beta_{k-1}}\|_2 \|\boldsymbol{\Lambda}_{\text{pos}}^{\beta_{k-1}} - \boldsymbol{\Lambda}_{\text{pos}}^{\beta_k}\|_2 \|\boldsymbol{\Sigma}_{\text{pos}}^{\beta_k}\|_2 \\
&= (\beta_k - \beta_{k-1}) \|\boldsymbol{\Sigma}_{\text{pos}}^{\beta_{k-1}}\|_2 \|\boldsymbol{\Lambda}_{\text{lld}}\|_2 \|\boldsymbol{\Sigma}_{\text{pos}}^{\beta_k}\|_2 \\
&= \frac{C_3}{K}
\end{aligned} \tag{28}$$

and

$$\begin{aligned}
\|\boldsymbol{\mu}_{\text{pos}}^{\beta_{k-1}} - \boldsymbol{\mu}_{\text{pos}}^{\beta_k}\|_2 &= \|\boldsymbol{\Sigma}_{\text{pos}}^{\beta_{k-1}} \beta_{k-1} \boldsymbol{\Lambda}_{\text{lld}} \boldsymbol{\mu}_* - \boldsymbol{\Sigma}_{\text{pos}}^{\beta_k} \beta_k \boldsymbol{\Lambda}_{\text{lld}} \boldsymbol{\mu}_*\|_2 \\
&\leq \|(\beta_k \boldsymbol{\Sigma}_{\text{pos}}^{\beta_k} - \beta_{k-1} \boldsymbol{\Sigma}_{\text{pos}}^{\beta_{k-1}})\|_2 \|\boldsymbol{\Lambda}_{\text{lld}} \boldsymbol{\mu}_*\|_2 \\
&= \|\beta_k (\boldsymbol{\Sigma}_{\text{pos}}^{\beta_k} - \boldsymbol{\Sigma}_{\text{pos}}^{\beta_{k-1}}) + (\beta_k - \beta_{k-1}) \boldsymbol{\Sigma}_{\text{pos}}^{\beta_{k-1}}\|_2 \|\boldsymbol{\Lambda}_{\text{lld}} \boldsymbol{\mu}_*\|_2 \\
&\leq \left(\beta_k \|\boldsymbol{\Sigma}_{\text{pos}}^{\beta_k} - \boldsymbol{\Sigma}_{\text{pos}}^{\beta_{k-1}}\|_2 + (\beta_k - \beta_{k-1}) \|\boldsymbol{\Sigma}_{\text{pos}}^{\beta_{k-1}}\|_2\right) \|\boldsymbol{\Lambda}_{\text{lld}} \boldsymbol{\mu}_*\|_2 \\
&= \left(\beta_k \frac{C_3}{K} + (\beta_k - \beta_{k-1}) \|\boldsymbol{\Sigma}_{\text{pos}}^{\beta_{k-1}}\|_2\right) \|\boldsymbol{\Lambda}_{\text{lld}} \boldsymbol{\mu}_*\|_2 \\
&= \frac{C_1}{K}
\end{aligned} \tag{29}$$

We now begin the induction for $\|\boldsymbol{\mu}_{k-1} - \boldsymbol{\mu}_{\text{pos}}^{\beta_k}\|_2$. For $k = 1$, we have $\boldsymbol{\mu}_0 = \boldsymbol{\mu}_p$ and obtain

$$\|\boldsymbol{\mu}_0 - \boldsymbol{\mu}_{\text{pos}}^{\beta_1}\|_2 = \|\boldsymbol{\mu}_{\text{pos}}^{\beta_0} - \boldsymbol{\mu}_{\text{pos}}^{\beta_1}\|_2 = \mathcal{O}(K^{-1}). \tag{30}$$

Next, we assume $\|\boldsymbol{\mu}_{k-1} - \boldsymbol{\mu}_{pos}^{\beta_k}\| = C_2 K^{2c-1} = \mathcal{O}(K^{2c-1})$ holds for $k \geq 1$. Subtracting $\boldsymbol{\mu}_{\text{pos}}^{\beta_k}$ from both sides of (73) yields

$$\boldsymbol{\mu}_k - \boldsymbol{\mu}_{\text{pos}}^{\beta_k} = \left(\mathbf{I} - \frac{\eta_k^2}{2} \boldsymbol{\Lambda}_{\text{pos}}^{\beta_k}\right) (\boldsymbol{\mu}_{k-1} - \boldsymbol{\mu}_{\text{pos}}^{\beta_k}). \tag{31}$$

As $\boldsymbol{\Lambda}_{\text{pos}}^{\beta_k} \succeq \boldsymbol{\Lambda}_p \succeq \mathbf{I}$ by construction, we have

$$\begin{aligned}
\|\boldsymbol{\mu}_k - \boldsymbol{\mu}_{\text{pos}}^{\beta_{k+1}}\|_2 &\leq \|\boldsymbol{\mu}_k - \boldsymbol{\mu}_{\text{pos}}^{\beta_k}\|_2 + \|\boldsymbol{\mu}_{\text{pos}}^{\beta_k} - \boldsymbol{\mu}_{\text{pos}}^{\beta_{k+1}}\|_2 \\
&\leq (1 - \frac{\eta^2}{2}) \|\boldsymbol{\mu}_{k-1} - \boldsymbol{\mu}_{\text{pos}}^{\beta_k}\|_2 + \frac{C_1}{K} \\
&= C_2 K^{2c-1} - \frac{a^2 C_2}{2} K^{-1} + C_1 K^{-1}
\end{aligned} \tag{32}$$

We have the flexibility to choose $a$ such that $a^2 C_2 \geq 2 C_1$, hence we have $\|\boldsymbol{\mu}_k - \boldsymbol{\mu}_{\text{pos}}^{\beta_{k+1}}\|_2 \leq C_2 K^{2c-1}$. This completes the proof for $\|\boldsymbol{\mu}_{k-1} - \boldsymbol{\mu}_{\text{pos}}^{\beta_k}\|_2$.

Now, we bound $\|\boldsymbol{\Lambda}_{k-1} - \boldsymbol{\Lambda}_{\text{pos}}^{\beta_k}\|_2$. It suffices to prove that $\|\boldsymbol{\Sigma}_{k-1} - \boldsymbol{\Sigma}_{\text{pos}}^{\beta_k}\|_2 = \mathcal{O}(K^{2c-1})$ because $\|\boldsymbol{\Lambda}_{k-1} - \boldsymbol{\Lambda}_{\text{pos}}^{\beta_k}\|_2 \leq \|\boldsymbol{\Lambda}_{k-1}\|_2 \|\boldsymbol{\Sigma}_{k-1} - \boldsymbol{\Sigma}_{\text{pos}}^{\beta_k}\|_2 \|\boldsymbol{\Lambda}_{\text{pos}}^{\beta_k}\|_2$. For $k = 1$, we have $\boldsymbol{\Sigma}_0 = \boldsymbol{\Sigma}_p$ and obtain

$$\|\boldsymbol{\Sigma}_0 - \boldsymbol{\Sigma}_{\text{pos}}^{\beta_1}\|_2 = \|\boldsymbol{\Sigma}_{\text{pos}}^{\beta_0} - \boldsymbol{\Sigma}_{\text{pos}}^{\beta_1}\|_2 = \mathcal{O}(K^{-1}) \tag{33}$$

by (28). Next, we assume $\|\boldsymbol{\Sigma}_{k-1} - \boldsymbol{\Sigma}_{\text{pos}}^{\beta_k}\|_2 \leq C_4 K^{2c-1}$ for $k \geq 1$. Subtracting $\boldsymbol{\Sigma}_{\text{pos}}^{\beta_{k+1}}$ from both sides of (75) yields

$$\boldsymbol{\Sigma}_k - \boldsymbol{\Sigma}_{\text{pos}}^{\beta_{k+1}} = \left(\mathbf{I} - \frac{\eta^2}{2}\boldsymbol{\Lambda}_{\text{pos}}^{\beta_k}\right)(\boldsymbol{\Sigma}_{k-1} - \boldsymbol{\Sigma}_{\text{pos}}^{\beta_k})\left(\mathbf{I} - \frac{\eta^2}{2}\boldsymbol{\Lambda}_{\text{pos}}^{\beta_k}\right)$$
$$+ \boldsymbol{\Sigma}_{\text{pos}}^{\beta_k} - \boldsymbol{\Sigma}_{\text{pos}}^{\beta_{k+1}} - \frac{\eta^4}{4}\boldsymbol{\Lambda}_{\text{pos}}^{\beta_k} + \frac{\eta^6}{16}(\boldsymbol{\Lambda}_{\text{pos}}^{\beta_k})^2. \tag{34}$$

By invoking $\boldsymbol{\Lambda}_{\text{pos}}^{\beta_k} \succeq \mathbf{I}$, we have

$$\|\boldsymbol{\Sigma}_k - \boldsymbol{\Sigma}_{\text{pos}}^{\beta_{k+1}}\|_2 \leq \left(1 - \frac{\eta^2}{2}\right)^2 \|\boldsymbol{\Sigma}_{k-1} - \boldsymbol{\Sigma}_{\text{pos}}^{\beta_k}\|_2 + \|\boldsymbol{\Sigma}_{\text{pos}}^{\beta_k} - \boldsymbol{\Sigma}_{\text{pos}}^{\beta_{k+1}}\|_2 + \frac{\eta^4}{4}\left\|\boldsymbol{\Lambda}_{\text{pos}}^{\beta_k} - \frac{\eta^2}{4}(\boldsymbol{\Lambda}_{\text{pos}}^{\beta_k})^2\right\|_2$$

$$\leq C_4 K^{2c-1} - a^2 C_4 K^{-1} + \frac{a^4 C_4}{4}K^{-2c-1} + C_3 K^{-1} + a^4 C_5 K^{-4c}. \tag{35}$$

To finish the proof, we can choose $a$ and $C_4$ to ensure $a^2 C_4 - C_3 - a^4 C_5 - \frac{a^4 C_4}{4} > 0$. $\qquad \square$

**Theorem 1.** *Given equally spaced $\beta_k$, running DAIS with $\gamma = 0$ and $\eta \sim \frac{1}{K^c}$ where $c \geq \frac{1}{4}$ yields*

$$\log p(\mathcal{D}) - \mathcal{L}_{DAIS} = \mathcal{O}(K^{2c-1}).$$

*With $c = \frac{1}{4}$, we have the convergence rate $\mathcal{O}(1/\sqrt{K})$.*

*Proof.* By Lemma 1, we have the first two terms in (17) upper bounded as follows,

$$\left|\frac{1}{2}\|\boldsymbol{\mu}_K - \boldsymbol{\mu}_{\text{pos}}\|_{\boldsymbol{\Lambda}_{\text{pos}}}^2 + \frac{1}{2}\text{Tr}(\boldsymbol{\Lambda}_{\text{pos}}\boldsymbol{\Sigma}_K) - \frac{d}{2}\right| \leq \frac{1}{2}\|\boldsymbol{\mu}_K - \boldsymbol{\mu}_{\text{pos}}\|_{\boldsymbol{\Lambda}_{\text{pos}}}^2 + \frac{1}{2}|\text{Tr}(\boldsymbol{\Lambda}_{\text{pos}}(\boldsymbol{\Sigma}_K - \boldsymbol{\Sigma}_{\text{pos}}))|$$
$$= \mathcal{O}(K^{4c-2}) + \mathcal{O}(K^{2c-1}) = \mathcal{O}(K^{2c-1}) \tag{36}$$

where we use the identity that $|\text{Tr}(\mathbf{AB})| \leq d\|\mathbf{A}\|_2\|\mathbf{B}\|_2$.

Now, it suffices to show the term ③ vanishes as $K \to \infty$. To put it differently, we only need to show that $\lim_{K \to \infty} \mathbb{E}_q\left[\sum_{k=1}^K \log \frac{\pi(\hat{\mathbf{v}}_k)}{\pi(\mathbf{v}_{k-1})}\right] = \frac{1}{2}\log\frac{|\boldsymbol{\Sigma}_{\text{pos}}|}{|\boldsymbol{\Sigma}_p|}$. To this end, we could first simplify $\mathbb{E}_q\left[\sum_{k=1}^K \log \frac{\pi(\hat{\mathbf{v}}_k)}{\pi(\mathbf{v}_{k-1})}\right]$ as we know $\hat{\mathbf{v}}_k$ follows a Gaussian distribution $\mathcal{N}(\boldsymbol{\mu}_k^{\mathbf{v}}, \boldsymbol{\Sigma}_k^{\mathbf{v}})$.

$$\mathbb{E}_q\left[\sum_{k=1}^K \log \frac{\pi(\hat{\mathbf{v}}_k)}{\pi(\mathbf{v}_{k-1})}\right] = \sum_{k=1}^K \left[-\frac{1}{2}\|\boldsymbol{\mu}_k^{\mathbf{v}}\|_2^2 - \frac{1}{2}\text{Tr}(\boldsymbol{\Sigma}_k^{\mathbf{v}}) + \frac{d}{2}\right] \tag{37}$$

According to (16), we have $\boldsymbol{\mu}_k^{\mathbf{v}} = -\eta\boldsymbol{\Lambda}_{\text{pos}}^{\beta_k}(\boldsymbol{\mu}_k - \boldsymbol{\mu}_{\text{pos}}^{\beta_k})$. Then by Lemma 1, we obtain $\|\boldsymbol{\mu}_k^{\mathbf{v}}\|_2^2 = \mathcal{O}(K^{2c-2})$ and $\sum_{k=1}^K[\frac{1}{2}\|\boldsymbol{\mu}_k^{\mathbf{v}}\|_2^2] = \mathcal{O}(K^{2c-1})$. Again, this term would vanish if we choose $c < \frac{1}{2}$. From (77), we can write $\text{Tr}(\boldsymbol{\Sigma}_k^{\mathbf{v}})$ as

$$\text{Tr}(\boldsymbol{\Sigma}_k^{\mathbf{v}}) = \text{Tr}\left(\eta^2\boldsymbol{\Lambda}_{\text{pos}}^{\beta_k}\boldsymbol{\Sigma}_{k-1}\boldsymbol{\Lambda}_{\text{pos}}^{\beta_k} + (\mathbf{I} - \frac{\eta^2}{2}\boldsymbol{\Lambda}_{\text{pos}}^{\beta_k})^2\right)$$
$$= \eta^2\text{Tr}\left(\boldsymbol{\Lambda}_{\text{pos}}^{\beta_k}\boldsymbol{\Sigma}_{k-1}(\boldsymbol{\Lambda}_{\text{pos}}^{\beta_k} - \boldsymbol{\Lambda}_{k-1})\right) + d + \frac{\eta^4}{4}\text{Tr}\left((\boldsymbol{\Lambda}_{\text{pos}}^{\beta_k})^2\right) \tag{38}$$

In addition, we have the recurrence for $\boldsymbol{\Sigma}_k$:

$$\boldsymbol{\Sigma}_k = \left(\mathbf{I} - \frac{1}{2}\eta^2\boldsymbol{\Lambda}_{\text{pos}}^{\beta_k}\right)\boldsymbol{\Sigma}_{k-1}\left(\mathbf{I} - \frac{1}{2}\eta^2\boldsymbol{\Lambda}_{\text{pos}}^{\beta_k}\right) + \eta^2\left(\mathbf{I} - \frac{1}{2}\eta^2\boldsymbol{\Lambda}_{\text{pos}}^{\beta_k} + \frac{\eta^4}{16}(\boldsymbol{\Lambda}_{\text{pos}}^{\beta_k})^2\right) \tag{39}$$

This immediately leads to

$$\text{Tr}\left((\mathbf{\Sigma}_k - \mathbf{\Sigma}_{k-1})\mathbf{\Lambda}_{\text{pos}}^{\beta_k}\right) = -\eta^2 \text{Tr}\left(\mathbf{\Lambda}_{\text{pos}}^{\beta_k}\mathbf{\Sigma}_{k-1}(\mathbf{\Lambda}_{\text{pos}}^{\beta_k} - \mathbf{\Lambda}_{k-1})\right)$$
$$+ \frac{\eta^4}{4}\text{Tr}\left(\mathbf{\Lambda}_{\text{pos}}^{\beta_k}\mathbf{\Sigma}_{k-1}(\mathbf{\Lambda}_{\text{pos}}^{\beta_k})^2\right) - \frac{\eta^4}{2}\text{Tr}\left((\mathbf{\Lambda}_{\text{pos}}^{\beta_k})^2\right) + \frac{\eta^6}{16}\text{Tr}\left((\mathbf{\Lambda}_{\text{pos}}^{\beta_k})^3\right) \tag{40}$$

where we used the identity $\text{Tr}(\mathbf{AB}) = \text{Tr}(\mathbf{BA})$. Plugging (40) back into (38), we have

$$\text{Tr}(\mathbf{\Sigma}_k^{\mathbf{v}}) = d - \text{Tr}\left((\mathbf{\Sigma}_k - \mathbf{\Sigma}_{k-1})\mathbf{\Lambda}_{\text{pos}}^{\beta_k}\right) + \frac{\eta^4}{4}\text{Tr}\left((\mathbf{\Lambda}_{\text{pos}}^{\beta_k} - \mathbf{\Lambda}_{k-1})\mathbf{\Sigma}_{k-1}(\mathbf{\Lambda}_{\text{pos}}^{\beta_k})^2\right) + \frac{\eta^6}{16}\text{Tr}\left((\mathbf{\Lambda}_{\text{pos}}^{\beta_k})^3\right) \tag{41}$$

Further, if we sum over all timesteps, we get

$$\sum_{k=1}^{K}\left[-\frac{1}{2}\text{Tr}(\mathbf{\Sigma}_k^{\mathbf{v}}) + \frac{d}{2}\right] =$$
$$\underbrace{\frac{1}{2}\sum_{k=1}^{K}\left[\text{Tr}\left((\mathbf{\Sigma}_k - \mathbf{\Sigma}_{k-1})\mathbf{\Lambda}_{k-1}\right)\right]}_{=\frac{1}{2}\log\frac{|\mathbf{\Sigma}_K|}{|\mathbf{\Sigma}_p|}+\mathcal{O}(K^{-1})} + \underbrace{\frac{1}{2}\sum_{k=1}^{K}\left[\text{Tr}\left((\mathbf{\Sigma}_k - \mathbf{\Sigma}_{k-1})(\mathbf{\Lambda}_{\text{pos}}^{\beta_k} - \mathbf{\Lambda}_{k-1})\right)\right]}_{=\mathcal{O}(K^{2c-1})}$$
$$\underbrace{-\frac{\eta^4}{8}\sum_{k=1}^{K}\left[\text{Tr}\left((\mathbf{\Lambda}_{\text{pos}}^{\beta_k} - \mathbf{\Lambda}_{k-1})\mathbf{\Sigma}_{k-1}(\mathbf{\Lambda}_{\text{pos}}^{\beta_k})^2\right)\right]}_{=\mathcal{O}(K^{-2c})} \underbrace{-\frac{\eta^6}{32}\sum_{k=1}^{K}\left[\text{Tr}\left((\mathbf{\Lambda}_{\text{pos}}^{\beta_k})^3\right)\right]}_{=\mathcal{O}(K^{-6c+1})} \tag{42}$$

For the first term, we used the Riemann sum approximation for the integral

$$\int_{\mathbf{\Sigma}_p}^{\mathbf{\Sigma}_K}\text{Tr}(\mathbf{\Sigma}^{-1}d\mathbf{\Sigma}) = \int_{\mathbf{\Sigma}_p}^{\mathbf{\Sigma}_K}d\log|\mathbf{\Sigma}| = \log|\mathbf{\Sigma}_K| - \log|\mathbf{\Sigma}_p|. \tag{43}$$

Importantly, the integral is independent of the path $\mathbf{\Sigma}(t)$ and we could choose the path to go through all $\mathbf{\Sigma}_k$. By the same argument of Riemann sum, the approximation error is bounded by $\mathcal{O}\left(\sum_{k=1}^{K}\|\mathbf{\Sigma}_k - \mathbf{\Sigma}_{k-1}\|_2^2\right)$. By (75), one can show that $\|\mathbf{\Sigma}_k - \mathbf{\Sigma}_{k-1}\|_2 = \mathcal{O}(K^{-1})$, so we have the approximation error $\mathcal{O}(K^{-1})$. For the second term, we used that fact that $|\text{Tr}(\mathbf{AB})| \leq d\|\mathbf{A}\|_2\|\mathbf{B}\|_2$. Therefore, the whole term will converge to $\frac{1}{2}\log\frac{|\mathbf{\Sigma}_K|}{|\mathbf{\Sigma}_p|}$ if $\frac{1}{4} \leq c < \frac{1}{2}$. Finally, by Lemma 1, we have

$$\log\frac{|\mathbf{\Sigma}_K|}{|\mathbf{\Sigma}_p|} = \log\frac{|\mathbf{\Sigma}_{\text{pos}}|}{|\mathbf{\Sigma}_p|} + \log\frac{|\mathbf{\Lambda}_{\text{pos}}|}{|\mathbf{\Lambda}_K|} = \log\frac{|\mathbf{\Sigma}_{\text{pos}}|}{|\mathbf{\Sigma}_p|} + \mathcal{O}(K^{2c-1}) \tag{44}$$

Hence, we have ③ $= \mathcal{O}(K^{2c-1})$ if $c \geq \frac{1}{4}$. Therefore, we have

$$\log p(\mathcal{D}) - \mathcal{L}_{\text{DAIS}} = ① + ② + ③ = \mathcal{O}(K^{2c-1}). \tag{45}$$

This completes the proof. $\qquad\square$

**Theorem 2.** *For stochastic DAIS with full momentum refreshment ($\gamma = 0$ in Algorithm 1) and any stepsize scheme, we have*

$$\liminf_{K\to\infty}|\log p(\mathcal{D}) - \mathcal{L}_{DAIS}| > 0. \tag{20}$$

*Proof.* Recall the suboptimality gap

$$\log p(\mathcal{D}) - \mathcal{L}_{\text{DAIS}} =$$
$$\underbrace{\frac{1}{2}\|\boldsymbol{\mu}_K - \boldsymbol{\mu}_{\text{pos}}\|_{\mathbf{\Lambda}_{\text{pos}}}^2}_{①} + \underbrace{\frac{1}{2}\text{Tr}(\mathbf{\Lambda}_{\text{pos}}\mathbf{\Sigma}_K) - \frac{d}{2}}_{②} + \underbrace{\frac{1}{2}\log\frac{|\mathbf{\Sigma}_{\text{pos}}|}{|\mathbf{\Sigma}_p|} - \mathbb{E}_q\left[\sum_{k=1}^{K}\log\frac{\pi(\hat{\mathbf{v}}_k)}{\pi(\mathbf{v}_{k-1})}\right]}_{③}. \tag{46}$$

First, we note that ① would stay unchanged for DAIS with stochastic gradient because $\boldsymbol{\mu}_K$ is independent of gradient noise. Second, the term ② would only become larger if we use stochastic

gradient becuase more noise is injected into the system. It is easy for one to show that $\hat{\boldsymbol{\Sigma}}_k \succeq \boldsymbol{\Sigma}_k$ where $\hat{\boldsymbol{\Sigma}}_k$ is the covariance of $\boldsymbol{\theta}_k$ when stochastic gradient is used ($\boldsymbol{\Sigma}_k$ is the covariance of $\boldsymbol{\theta}_k$ in the full-batch setting). We will prove this by induction. By (19) and (16), we have

$$\hat{\boldsymbol{\Sigma}}_k = (\mathbf{I} - \frac{\eta^2}{2}\boldsymbol{\Lambda}_{\text{pos}}^{\beta_k})\hat{\boldsymbol{\Sigma}}_{k-1}(\mathbf{I} - \frac{\eta^2}{2}\boldsymbol{\Lambda}_{\text{pos}}^{\beta_k}) + \eta^2(\mathbf{I} - \frac{\eta^2}{2}\boldsymbol{\Lambda}_{\text{pos}}^{\beta_k} + \frac{\eta^4}{16}(\boldsymbol{\Lambda}_{\text{pos}}^{\beta_k})^2) + \frac{\eta^4}{4}\boldsymbol{\Sigma}_{\boldsymbol{\varepsilon}} \quad (47)$$

$$\boldsymbol{\Sigma}_k = (\mathbf{I} - \frac{\eta^2}{2}\boldsymbol{\Lambda}_{\text{pos}}^{\beta_k})\boldsymbol{\Sigma}_{k-1}(\mathbf{I} - \frac{\eta^2}{2}\boldsymbol{\Lambda}_{\text{pos}}^{\beta_k}) + \eta^2(\mathbf{I} - \frac{\eta^2}{2}\boldsymbol{\Lambda}_{\text{pos}}^{\beta_k} + \frac{\eta^4}{16}(\boldsymbol{\Lambda}_{\text{pos}}^{\beta_k})^2) \quad (48)$$

At $k = 1$, we know $\hat{\boldsymbol{\Sigma}}_0 = \boldsymbol{\Sigma}_0 = \boldsymbol{\Sigma}_p$, so we retrieve the base case $\hat{\boldsymbol{\Sigma}}_1 - \boldsymbol{\Sigma}_1 = \frac{\eta^4}{4}\boldsymbol{\Sigma}_{\boldsymbol{\varepsilon}} \succeq 0$. Now assume $\hat{\boldsymbol{\Sigma}}_{k-1} \succeq \boldsymbol{\Sigma}_{k-1}$, then

$$\hat{\boldsymbol{\Sigma}}_k - \boldsymbol{\Sigma}_k = \underbrace{(\mathbf{I} - \frac{\eta^2}{2}\boldsymbol{\Lambda}_{\text{pos}}^{\beta_k})(\hat{\boldsymbol{\Sigma}}_{k-1} - \boldsymbol{\Sigma}_{k-1})(\mathbf{I} - \frac{\eta^2}{2}\boldsymbol{\Lambda}_{\text{pos}}^{\beta_k})}_{\succeq 0} + \frac{\eta^4}{4}\boldsymbol{\Sigma}_{\boldsymbol{\varepsilon}} \succeq 0 \quad (49)$$

which completes the induction.

Therefore, we only need to compare ③ of the stochastic gradient variant to its deterministic counterpart. It suffices to show that ③ becomes larger once we use stochastic gradient for the updates. To this end, we let $\tilde{\boldsymbol{\Sigma}}_k^{\mathbf{v}}$ to be the covariance of $\hat{\mathbf{v}}_k$ and have the following recursion (by (19))

$$\tilde{\boldsymbol{\Sigma}}_k^{\mathbf{v}} = \eta_k^2\boldsymbol{\Lambda}_{\text{pos}}^{\beta_k}\hat{\boldsymbol{\Sigma}}_{k-1}\boldsymbol{\Lambda}_{\text{pos}}^{\beta_k} + (\mathbf{I} - \frac{\eta_k^2}{2}\boldsymbol{\Lambda}_{\text{pos}}^{\beta_k})^2 + \eta_k^2\boldsymbol{\Sigma}_{\boldsymbol{\varepsilon}}. \quad (50)$$

For notational convenience, we let $\hat{\boldsymbol{\Sigma}}_k^{\mathbf{v}} \triangleq \eta_k^2\boldsymbol{\Lambda}_{\text{pos}}^{\beta_k}\hat{\boldsymbol{\Sigma}}_{k-1}\boldsymbol{\Lambda}_{\text{pos}}^{\beta_k} + (\mathbf{I} - \frac{\eta_k^2}{2}\boldsymbol{\Lambda}_{\text{pos}}^{\beta_k})^2$. Here, we used $\tilde{\boldsymbol{\Sigma}}_k^{\mathbf{v}}, \hat{\boldsymbol{\Sigma}}_k^{\mathbf{v}}$ and $\hat{\boldsymbol{\Sigma}}_k$ to avoid confusion with $\boldsymbol{\Sigma}_k^{\mathbf{v}}$ and $\boldsymbol{\Sigma}_k$ in the full-batch setting. For convenience, we further assume $\boldsymbol{\varepsilon}$ is Gaussian (appealing to the central limit theorem) and get

$$\mathbb{E}_q\left[\sum_{k=1}^K \log\frac{\pi(\hat{\mathbf{v}}_k)}{\pi(\mathbf{v}_{k-1})}\right] = \sum_{k=1}^K\left[-\frac{1}{2}\|\boldsymbol{\mu}_k^{\mathbf{v}}\|_2^2 - \frac{1}{2}\text{Tr}(\hat{\boldsymbol{\Sigma}}_k^{\mathbf{v}}) + \frac{d}{2}\right] - \sum_{k=1}^K\left[\frac{1}{2}\eta_k^2\text{Tr}(\boldsymbol{\Sigma}_{\boldsymbol{\varepsilon}})\right]. \quad (51)$$

Importantly, we notice that $\hat{\boldsymbol{\Sigma}}_k^{\mathbf{v}} \succeq \boldsymbol{\Sigma}_k^{\mathbf{v}}$ because $\hat{\boldsymbol{\Sigma}}_k \succeq \boldsymbol{\Sigma}_k$. So the suboptimality gap (46) increases at least by $\sum_{k=1}^K\left[\frac{1}{2}\eta_k^2\text{Tr}(\boldsymbol{\Sigma}_{\boldsymbol{\varepsilon}})\right]$ with stochastic gradient update in DAIS. This immediately implies the necessary condition of DAIS being consistent is

$$\lim_{K\to\infty}\sum_{k=1}^K \eta_k^2 = 0, \quad (52)$$

where we assume $\text{Tr}(\boldsymbol{\Sigma}_{\boldsymbol{\varepsilon}}) > 0$. We now show that for $\boldsymbol{\mu}_K$ to converge to posterior mean $\boldsymbol{\mu}_{\text{pos}}$ (so that ① vanishes), a major requirement is $\lim_{K\to\infty}\sum_{k=1}^K \eta_k^2 = \infty$. To prove that, we observe that the mean of $\boldsymbol{\theta}_k$ evolves as follows:

$$\boldsymbol{\mu}_k - \boldsymbol{\mu}_{\text{pos}}^{\beta_k} \leftarrow \left(\mathbf{I} - \frac{\eta_k^2}{2}\boldsymbol{\Lambda}_{\text{pos}}^{\beta_k}\right)(\boldsymbol{\mu}_{k-1} - \boldsymbol{\mu}_{\text{pos}}^{\beta_k}) \quad (53)$$

Since $\eta_k$ is $o(1)$ by (52) and we know $\boldsymbol{\Lambda}_{\text{pos}}^{\beta_k}$ is upper bounded by $C\mathbf{I}$ for some constant $C$. One can show the following by induction:

$$\|\boldsymbol{\mu}_K - \boldsymbol{\mu}_{\text{pos}}\|_2 \geq \left\|\left(\prod_{k=1}^K\left(1 - \frac{C\eta_k^2}{2}\right)\right)(\boldsymbol{\mu}_0 - \boldsymbol{\mu}_{\text{pos}})\right\|_2 \quad (54)$$

For $\|\boldsymbol{\mu}_K - \boldsymbol{\mu}_{\text{pos}}\|_2 \to 0$ in the worst case, it requires the following to hold:

$$\lim_{K\to\infty}\prod_{k=1}^K\left(1 - \frac{C\eta_k^2}{2}\right) = 0. \quad (55)$$

This is equivalent to

$$\lim_{K\to\infty}\sum_{k=1}^K \log\left(1 - \frac{C\eta_k^2}{2}\right) \stackrel{\eta_k = o(1)}{\approx} -\lim_{K\to\infty}\sum_{k=1}^K \frac{C\eta_k^2}{2} = -\infty \quad (56)$$

This completes the proof. $\qquad\qquad\square$

# B   Other Derivations

## B.1   DAIS Update

Here, we derive the DAIS updates for the position and (pre-refreshment) momentum of the parameter particles for the Bayesian linear regression setting. Recall that the update from step $k-1$ to step $k$ takes the general form

$$\boldsymbol{\theta}_{k-\frac{1}{2}} = \boldsymbol{\theta}_{k-1} + \frac{\eta}{2}\mathbf{M}^{-1}\mathbf{v}_{k-1} \tag{57}$$

$$\hat{\mathbf{v}}_k = \mathbf{v}_{k-1} + \eta\nabla\log f_{\beta_k}(\boldsymbol{\theta}_{k-\frac{1}{2}}) \tag{58}$$

$$\boldsymbol{\theta}_k = \boldsymbol{\theta}_{k-\frac{1}{2}} + \frac{\eta}{2}\mathbf{M}^{-1}\hat{\mathbf{v}}_k \tag{59}$$

Under a geometric annealing scheme, the log annealed unnormalized posterior at step $k$ has the form

$$\begin{aligned}
\log f_{\beta_k}(\boldsymbol{\theta}) &= \log\left(p(\boldsymbol{\theta}|\mathcal{M})p(\mathcal{D}|\boldsymbol{\theta},\mathcal{M})^{\beta_k}\right) \\
&= -\frac{1}{2}(\boldsymbol{\theta}-\boldsymbol{\mu}_p)^\top\boldsymbol{\Lambda}_p(\boldsymbol{\theta}-\boldsymbol{\mu}_p) - \frac{\beta_k}{2\sigma^2}(\mathbf{y}-\mathbf{X}\boldsymbol{\theta})^\top(\mathbf{y}-\mathbf{X}\boldsymbol{\theta}) + C,
\end{aligned} \tag{60}$$

where $C$ comprises terms constant w.r.t. $\boldsymbol{\theta}$. The gradient of this is then

$$\nabla_{\boldsymbol{\theta}}\log f_{\beta_k}(\boldsymbol{\theta}) = -\boldsymbol{\Lambda}_p(\boldsymbol{\theta}-\boldsymbol{\mu}_p) + \frac{\beta_k}{\sigma^2}\mathbf{X}^\top(\mathbf{y}-\mathbf{X}\boldsymbol{\theta}). \tag{61}$$

Substituting, we have

$$\begin{aligned}
\nabla_{\boldsymbol{\theta}_{k-\frac{1}{2}}}\log f_{\beta_k}(\boldsymbol{\theta}_{k-\frac{1}{2}}) = &-\boldsymbol{\Lambda}_p\left(\boldsymbol{\theta}_{k-1} + \frac{\eta_k}{2}\mathbf{v}_{k-1} - \boldsymbol{\mu}_p\right) \\
&+ \frac{\beta_k}{\sigma^2}\mathbf{X}^\top\left(y - \mathbf{X}\left(\boldsymbol{\theta}_{k-1} + \frac{\eta_k}{2}\mathbf{v}_{k-1}\right)\right)
\end{aligned} \tag{62}$$

Define the likelihood precision, annealed posterior precision, and annealed posterior mean as

$$\boldsymbol{\Lambda}_{\text{lld}} = \frac{1}{\sigma^2}\mathbf{X}^\top\mathbf{X} \tag{63}$$

$$\boldsymbol{\Lambda}_{\text{pos}}^{\beta_k} = \boldsymbol{\Lambda}_p + \beta_k\boldsymbol{\Lambda}_{\text{lld}} = \boldsymbol{\Lambda}_p + \frac{\beta_k}{\sigma^2}\mathbf{X}^\top\mathbf{X} \tag{64}$$

$$\boldsymbol{\mu}_{\text{pos}}^{\beta_k} = \left(\boldsymbol{\Lambda}_{\text{pos}}^{\beta_k}\right)^{-1}\left(\boldsymbol{\Lambda}_p\boldsymbol{\mu}_p + \beta_k\boldsymbol{\Lambda}_{\text{lld}}\left(\mathbf{X}^\top\mathbf{X}\right)^{-1}\mathbf{X}^\top\mathbf{y}\right). \tag{65}$$

For the updated pre-refreshment momentum, we have

$$\begin{aligned}
\hat{\mathbf{v}}_k &= \mathbf{v}_{k-1} + \eta_k\nabla_{\boldsymbol{\theta}_{k-\frac{1}{2}}}\log f_{\beta_k}(\boldsymbol{\theta}_{k-\frac{1}{2}}) \\
&= \mathbf{v}_{k-1} - \eta_k\boldsymbol{\Lambda}_p\boldsymbol{\theta}_{k-1} - \frac{\eta_k^2}{2}\boldsymbol{\Lambda}_p\mathbf{v}_{k-1} + \eta_k\boldsymbol{\Lambda}_p\boldsymbol{\mu}_p \\
&\quad + \frac{\beta_k\eta_k}{\sigma^2}\mathbf{X}^\top\mathbf{y} - \frac{\beta_k\eta_k}{\sigma^2}\mathbf{X}^\top\mathbf{X}\boldsymbol{\theta}_{k-1} + \frac{\beta_k\eta_k^2}{2\sigma^2}\mathbf{X}^\top\mathbf{X}\mathbf{v}_{k-1} \\
&= \left(\mathbf{I} - \frac{\eta_k^2}{2}\left(\boldsymbol{\Lambda}_p + \frac{\beta_k}{\sigma^2}\mathbf{X}^\top\mathbf{X}\right)\right)\mathbf{v}_{k-1} - \eta_k\left(\boldsymbol{\Lambda}_p + \frac{\beta_k}{\sigma^2}\mathbf{X}^\top\mathbf{X}\right)\boldsymbol{\theta}_{k-1} \\
&\quad + \eta_k\left(\boldsymbol{\Lambda}_p\boldsymbol{\mu}_p + \frac{\beta_k}{\sigma^2}\mathbf{X}^\top\mathbf{y}\right) \\
&= \left(\mathbf{I} - \frac{\eta_k^2}{2}\boldsymbol{\Lambda}_{\text{pos}}^{\beta_k}\right)\mathbf{v}_{k-1} - \eta_k\boldsymbol{\Lambda}_{\text{pos}}^{\beta_k}\boldsymbol{\theta}_{k-1} + \eta_k\boldsymbol{\Lambda}_{\text{pos}}^{\beta_k}\boldsymbol{\mu}_{\text{pos}}^{\beta_k},
\end{aligned} \tag{66}$$

and for the updated position, we have

$$\begin{aligned}
\boldsymbol{\theta}_k &= \boldsymbol{\theta}_{k-\frac{1}{2}} + \frac{\eta_k}{2}\hat{\mathbf{v}}_k \\
&= \boldsymbol{\theta}_{k-1} + \frac{\eta_k}{2}\mathbf{v}_{k-1} + \frac{\eta_k}{2}\left(\left(\mathbf{I} - \frac{\eta_k^2}{2}\boldsymbol{\Lambda}_{\text{pos}}^{\beta_k}\right)\mathbf{v}_{k-1} - \eta_k\boldsymbol{\Lambda}_{\text{pos}}^{\beta_k}\boldsymbol{\theta}_{k-1} + \eta_k\boldsymbol{\Lambda}_{\text{pos}}^{\beta_k}\boldsymbol{\mu}_{\text{pos}}^{\beta_k}\right) \\
&= \left(\mathbf{I} - \frac{\eta_k^2}{2}\boldsymbol{\Lambda}_{\text{pos}}^{\beta_k}\right)\boldsymbol{\theta}_{k-1} + \left(\eta_k\mathbf{I} - \frac{\eta^3}{4}\boldsymbol{\Lambda}_{\text{pos}}^{\beta_k}\right)\mathbf{v}_{k-1} + \frac{\eta_k^2}{2}\boldsymbol{\Lambda}_{\text{pos}}^{\beta_k}\boldsymbol{\mu}_{\text{pos}}^{\beta_k}.
\end{aligned} \tag{67}$$

## B.2 Expected DAIS for Bayesian Linear Regression

In this section, we derive the gap between the exact log marginal likelihood and the lower bound given in expectation by DAIS.

If $\mathbf{y}$ is distributed as a Gaussian with mean $\mathbf{X}\boldsymbol{\theta}$ and covariance $\sigma^2 \mathbf{I}$, and $q(\boldsymbol{\theta})$ is the density of a random variable with mean $\boldsymbol{\mu}_q$ and covariance $\boldsymbol{\Sigma}_q$, then the expected log-likelihood is

$$
\begin{aligned}
\mathbb{E}_q \left[ \log p(\mathbf{y}|\mathbf{X}, \boldsymbol{\theta}) \right] &= \mathbb{E}_q \left[ -\frac{n}{2} \log(2\pi\sigma^2) - \frac{1}{2\sigma^2} (\mathbf{y} - \mathbf{X}\boldsymbol{\theta})^\top (\mathbf{y} - \mathbf{X}\boldsymbol{\theta}) \right] \\
&= \mathbb{E}_q \left[ -\frac{n}{2} \log(2\pi\sigma^2) - \frac{1}{2\sigma^2} \left( \mathbf{y}^\top \mathbf{y} - 2\boldsymbol{\theta}^\top \mathbf{X}^\top \mathbf{y} + \boldsymbol{\theta}^\top \mathbf{X}^\top \mathbf{X}\boldsymbol{\theta} \right) \right] \\
&= -\frac{n}{2} \log(2\pi\sigma^2) - \frac{1}{2\sigma^2} \left( \mathbf{y}^\top \mathbf{y} - 2\boldsymbol{\mu}_q^\top \mathbf{X}^\top \mathbf{y} + \text{Tr}(\mathbf{X}^\top \mathbf{X}\boldsymbol{\Sigma}_q) + \boldsymbol{\mu}_q \mathbf{X}^\top \mathbf{X}\boldsymbol{\mu}_q \right) \\
&= -\frac{n}{2} \log(2\pi\sigma^2) - \frac{1}{2\sigma^2} \left( \mathbf{y} - \mathbf{X}\boldsymbol{\mu}_q \right)^\top \left( \mathbf{y} - \mathbf{X}\boldsymbol{\mu}_q \right) - \frac{1}{2} \text{Tr}(\boldsymbol{\Lambda}_{\text{lld}} \boldsymbol{\Sigma}_q).
\end{aligned}
\tag{68}
$$

If $p(\boldsymbol{\theta})$ is the density of a Gaussian with mean $\boldsymbol{\mu}_p$ and covariance $\boldsymbol{\Sigma}_p$, and if $q(\boldsymbol{\theta})$ is the density of another random variable with mean $\boldsymbol{\mu}_q$ and covariance $\boldsymbol{\Sigma}_q$, then

$$
\begin{aligned}
\mathbb{E}_q \left[ \log p(\boldsymbol{\theta}) \right] &= \mathbb{E}_q \left[ -\frac{d}{2} \log(2\pi) - \frac{1}{2} \log |\boldsymbol{\Sigma}_p| - \frac{1}{2} (\boldsymbol{\theta} - \boldsymbol{\mu}_p)^\top \boldsymbol{\Lambda}_p (\boldsymbol{\theta} - \boldsymbol{\mu}_p) \right] \\
&= \mathbb{E}_q \left[ -\frac{d}{2} \log(2\pi) - \frac{1}{2} \log |\boldsymbol{\Sigma}_p| - \frac{1}{2} (\boldsymbol{\theta}^\top \boldsymbol{\Lambda}_p \boldsymbol{\theta} - 2\boldsymbol{\mu}_p^\top \boldsymbol{\Lambda}_p \boldsymbol{\theta} + \boldsymbol{\mu}_p^\top \boldsymbol{\Lambda}_p \boldsymbol{\mu}_p) \right] \\
&= -\frac{d}{2} \log(2\pi) - \frac{1}{2} \log |\boldsymbol{\Sigma}_p| - \frac{1}{2} \left( \text{Tr}(\boldsymbol{\Lambda}_p \boldsymbol{\Sigma}_q) + \boldsymbol{\mu}_q^\top \boldsymbol{\Lambda}_p \boldsymbol{\mu}_q - 2\boldsymbol{\mu}_p^\top \boldsymbol{\Lambda}_p \boldsymbol{\mu}_q + \boldsymbol{\mu}_p^\top \boldsymbol{\Lambda}_p \boldsymbol{\mu}_p \right) \\
&= -\frac{d}{2} \log(2\pi) - \frac{1}{2} \log |\boldsymbol{\Sigma}_p| - \frac{1}{2} \text{Tr}(\boldsymbol{\Lambda}_p \boldsymbol{\Sigma}_q) - \frac{1}{2} (\boldsymbol{\mu}_q - \boldsymbol{\mu}_p)^\top \boldsymbol{\Lambda}_p (\boldsymbol{\mu}_q - \boldsymbol{\mu}_p).
\end{aligned}
\tag{69}
$$

Given the above results, we now compute the DAIS lower bound:

$$
\begin{aligned}
\mathcal{L}_{\text{DAIS}} &= \mathbb{E}_{q_{\text{fwd}}} \left[ \log f_K(\boldsymbol{\theta}_K) - \log p_0(\boldsymbol{\theta}_0) + \sum_{k=1}^K \log \frac{\pi(\hat{\mathbf{v}}_k)}{\pi(\mathbf{v}_{k-1})} \right] \\
&= \mathbb{E}_{q_{\text{fwd}}} \left[ \log p(\mathcal{D}|\boldsymbol{\theta}_K) + \log p_0(\boldsymbol{\theta}_K) - \log p_0(\boldsymbol{\theta}_0) + \sum_{k=1}^K \log \frac{\pi(\hat{\mathbf{v}}_k)}{\pi(\mathbf{v}_{k-1})} \right] \\
&= \left( -\frac{n}{2} \log(2\pi\sigma^2) - \frac{1}{2\sigma^2} \mathbf{y}^\top \mathbf{y} + \frac{1}{\sigma^2} \boldsymbol{\mu}_K^\top \mathbf{X}^\top \mathbf{y} - \frac{1}{2} \text{Tr}(\boldsymbol{\Lambda}_{\text{lld}} \boldsymbol{\Sigma}_K) - \frac{1}{2} \boldsymbol{\mu}_K^\top \boldsymbol{\Lambda}_{\text{lld}} \boldsymbol{\mu}_K \right) \\
&\quad \left( -\frac{1}{2} \text{Tr}(\boldsymbol{\Lambda}_p \boldsymbol{\Sigma}_K) - \frac{1}{2} \boldsymbol{\mu}_K^\top \boldsymbol{\Lambda}_p \boldsymbol{\mu}_K + \boldsymbol{\mu}_p^\top \boldsymbol{\Lambda}_p \boldsymbol{\mu}_K - \frac{1}{2} \boldsymbol{\mu}_p^\top \boldsymbol{\Lambda}_p \boldsymbol{\mu}_p \right) + \frac{d}{2} + \mathbb{E}_{q_{\text{fwd}}} \left[ \sum_{k=1}^K \log \frac{\pi(\hat{\mathbf{v}}_k)}{\pi(\mathbf{v}_{k-1})} \right] \\
&= -\frac{n}{2} \log(2\pi\sigma^2) - \frac{1}{2\sigma^2} \mathbf{y}^\top \mathbf{y} + \frac{1}{\sigma^2} \boldsymbol{\mu}_K^\top \mathbf{X}^\top \mathbf{y} - \frac{1}{2} \text{Tr}(\boldsymbol{\Lambda}_{\text{pos}} \boldsymbol{\Sigma}_K) - \frac{1}{2} \boldsymbol{\mu}_K^\top \boldsymbol{\Lambda}_{\text{pos}} \boldsymbol{\mu}_K \\
&\quad + \boldsymbol{\mu}_p^\top \boldsymbol{\Lambda}_p \boldsymbol{\mu}_K - \frac{1}{2} \boldsymbol{\mu}_p^\top \boldsymbol{\Lambda}_p \boldsymbol{\mu}_p + \frac{d}{2} + \mathbb{E}_{q_{\text{fwd}}} \left[ \sum_{k=1}^K \log \frac{\pi(\hat{\mathbf{v}}_k)}{\pi(\mathbf{v}_{k-1})} \right]
\end{aligned}
\tag{70}
$$

For comparison, we compute the log marginal likelihood by completing the square and recognizing the normalization of a Gaussian density:

$$
\log p(\mathcal{D}) = \log \int p(\mathcal{D}|\boldsymbol{\theta})p_0(\boldsymbol{\theta})d\boldsymbol{\theta}
$$

$$
= \log \int (2\pi\sigma^2)^{-n/2} \exp\left(-\frac{1}{2}(\boldsymbol{\theta}-\boldsymbol{\mu}_*)^\top \boldsymbol{\Lambda}_{\text{lld}}(\boldsymbol{\theta}-\boldsymbol{\mu}_*) + \frac{1}{2}\boldsymbol{\mu}_*^\top \boldsymbol{\Lambda}_{\text{lld}}\boldsymbol{\mu}_* - \frac{1}{2\sigma^2}\mathbf{y}^\top \mathbf{y}\right)
$$

$$
(2\pi)^{-d/2}|\boldsymbol{\Sigma}_p|^{-1/2} \exp\left(-\frac{1}{2}(\boldsymbol{\theta}-\boldsymbol{\mu}_p)^\top \boldsymbol{\Lambda}_p(\boldsymbol{\theta}-\boldsymbol{\mu}_p)\right) d\boldsymbol{\theta}
$$

$$
= \log \int (2\pi\sigma^2)^{-n/2}(2\pi)^{-d/2}|\boldsymbol{\Sigma}_p|^{-1/2} \exp\left(-\frac{1}{2}(\boldsymbol{\theta}-\boldsymbol{\mu}_{\text{pos}})^\top \boldsymbol{\Lambda}_{\text{pos}}(\boldsymbol{\theta}-\boldsymbol{\mu}_{\text{pos}})\right.
$$

$$
\left. + \frac{1}{2}\boldsymbol{\mu}_{\text{pos}}^\top\boldsymbol{\Lambda}_{\text{pos}}\boldsymbol{\mu}_{\text{pos}} - \frac{1}{2\sigma^2}\mathbf{y}^\top\mathbf{y} - \frac{1}{2}\boldsymbol{\mu}_p^\top \boldsymbol{\Lambda}_p\boldsymbol{\mu}_p \right) d\boldsymbol{\theta}
$$

$$
= -\frac{n}{2}\log(2\pi\sigma^2) - \frac{d}{2}\cancel{\log(2\pi)} - \frac{1}{2}\log|\boldsymbol{\Sigma}_p| + \frac{1}{2}\boldsymbol{\mu}_{\text{pos}}^\top\boldsymbol{\Lambda}_{\text{pos}}\boldsymbol{\mu}_{\text{pos}} - \frac{1}{2\sigma^2}\mathbf{y}^\top\mathbf{y} - \frac{1}{2}\boldsymbol{\mu}_p^\top\boldsymbol{\Lambda}_p\boldsymbol{\mu}_p
$$

$$
\underbrace{+ \log \int \exp\left(-\frac{1}{2}(\boldsymbol{\theta}-\boldsymbol{\mu}_{\text{pos}})^\top\boldsymbol{\Lambda}_{\text{pos}}(\boldsymbol{\theta}-\boldsymbol{\mu}_{\text{pos}})\right)d\boldsymbol{\theta}}_{+\frac{d}{2}\cancel{\log(2\pi)}+\frac{1}{2}\log|\boldsymbol{\Sigma}_{\text{pos}}|}
$$

$$
= -\frac{n}{2}\log(2\pi\sigma^2) + \frac{1}{2}\log\frac{|\boldsymbol{\Sigma}_{\text{pos}}|}{|\boldsymbol{\Sigma}_p|} + \frac{1}{2}\boldsymbol{\mu}_{\text{pos}}^\top\boldsymbol{\Lambda}_{\text{pos}}\boldsymbol{\mu}_{\text{pos}} - \frac{1}{2\sigma^2}\mathbf{y}^\top\mathbf{y} - \frac{1}{2}\boldsymbol{\mu}_p^\top\boldsymbol{\Lambda}_p\boldsymbol{\mu}_p
$$

$$
\tag{71}
$$

The gap is thus

$$
\log p(\mathcal{D}) - \mathcal{L}_{\text{DAIS}} = \cancel{-\frac{n}{2}\log(2\pi\sigma^2)} + \frac{1}{2}\log\frac{|\boldsymbol{\Sigma}_{\text{pos}}|}{|\boldsymbol{\Sigma}_p|} + \frac{1}{2}\boldsymbol{\mu}_{\text{pos}}^\top\boldsymbol{\Lambda}_{\text{pos}}\boldsymbol{\mu}_{\text{pos}} - \cancel{\frac{1}{2\sigma^2}\mathbf{y}^\top\mathbf{y}}
$$

$$
\cancel{-\frac{1}{2}\boldsymbol{\mu}_p^\top\boldsymbol{\Lambda}_p\boldsymbol{\mu}_p} + \cancel{\frac{n}{2}\log(2\pi\sigma^2)} + \cancel{\frac{1}{2\sigma^2}\mathbf{y}^\top\mathbf{y}} - \frac{1}{\sigma^2}\boldsymbol{\mu}_K^\top\mathbf{X}^\top\mathbf{y} + \frac{1}{2}\text{Tr}(\boldsymbol{\Lambda}_{\text{pos}}\boldsymbol{\Sigma}_K)
$$

$$
+ \frac{1}{2}\boldsymbol{\mu}_K^\top\boldsymbol{\Lambda}_{\text{pos}}\boldsymbol{\mu}_K - \boldsymbol{\mu}_p^\top\boldsymbol{\Lambda}_p\boldsymbol{\mu}_K + \cancel{\frac{1}{2}\boldsymbol{\mu}_p^\top\boldsymbol{\Lambda}_p\boldsymbol{\mu}_p} - \frac{d}{2} - \mathbb{E}_{q_{\text{fwd}}}\left[\sum_{k=1}^{K}\log\frac{\pi(\hat{\mathbf{v}}_k)}{\pi(\mathbf{v}_{k-1})}\right]
$$

$$
= \underbrace{\frac{1}{2}\boldsymbol{\mu}_K^\top\boldsymbol{\Lambda}_{\text{pos}}\boldsymbol{\mu}_K - \frac{1}{\sigma^2}\boldsymbol{\mu}_K^\top\mathbf{X}^\top\mathbf{y} - \boldsymbol{\mu}_p^\top\boldsymbol{\Lambda}_p\boldsymbol{\mu}_K + \frac{1}{2}\boldsymbol{\mu}_{\text{pos}}^\top\boldsymbol{\Lambda}_{\text{pos}}\boldsymbol{\mu}_{\text{pos}}}_{\frac{1}{2}\|\boldsymbol{\mu}_K - \boldsymbol{\mu}_{\text{pos}}\|_{\boldsymbol{\Lambda}_{\text{pos}}}^2}
$$

$$
+ \frac{1}{2}\text{Tr}(\boldsymbol{\Lambda}_{\text{pos}}\boldsymbol{\Sigma}_K) - \frac{d}{2} + \frac{1}{2}\log\frac{|\boldsymbol{\Sigma}_{\text{pos}}|}{|\boldsymbol{\Sigma}_p|} - \mathbb{E}_{q_{\text{fwd}}}\left[\sum_{k=1}^{K}\log\frac{\pi(\hat{\mathbf{v}}_k)}{\pi(\mathbf{v}_{k-1})}\right]
$$

$$
\tag{72}
$$

## B.3 DAIS Update under Full Momentum Refreshment

The following quantities are used in our convergence analysis for DAIS, which assumes full momentum refreshment, i.e. that $\gamma = 0$.

For the mean of the updated position, we have

$$
\boldsymbol{\mu}_k = \mathbb{E}_{q_{\text{fwd}}}\left[\left(\mathbf{I} - \frac{\eta_k^2}{2}\boldsymbol{\Lambda}_{\text{pos}}^{\beta_k}\right)\boldsymbol{\theta}_{k-1} + \left(\eta_k\mathbf{I} - \frac{\eta^3}{4}\boldsymbol{\Lambda}_{\text{pos}}^{\beta_k}\right)\mathbf{v}_{k-1} + \frac{\eta_k^2}{2}\boldsymbol{\Lambda}_{\text{pos}}^{\beta_k}\boldsymbol{\mu}_{\text{pos}}^{\beta_k}\right]
$$

$$
= \left(\mathbf{I} - \frac{\eta_k^2}{2}\boldsymbol{\Lambda}_{\text{pos}}^{\beta_k}\right)\boldsymbol{\mu}_{k-1} + \frac{\eta_k^2}{2}\boldsymbol{\Lambda}_{\text{pos}}^{\beta_k}\boldsymbol{\mu}_{\text{pos}}^{\beta_k}
$$

$$
\tag{73}
$$

as $\mathbb{E}_{q_{\text{fwd}}}[\mathbf{v}_{k-1}] = \mathbf{0}$ under full momentum refreshment.

For the covariance of the updated position, we have

$$
\boldsymbol{\theta}_k - \boldsymbol{\mu}_k = \left(\mathbf{I} - \frac{\eta^2}{2}\boldsymbol{\Lambda}_{\text{pos}}^{\beta_k}\right)(\boldsymbol{\theta}_{k-1} - \boldsymbol{\mu}_{k-1}) + \left(\eta\mathbf{I} - \frac{\eta^3}{4}\boldsymbol{\Lambda}_{\text{pos}}^{\beta_k}\right)\mathbf{v}_{k-1}
\tag{74}
$$

and so

$$
\begin{aligned}
\boldsymbol{\Sigma}_k &= \mathbb{E}_{q_{\text{fwd}}}[(\boldsymbol{\theta}_k - \boldsymbol{\mu}_k)(\boldsymbol{\theta}_k - \boldsymbol{\mu}_k)^\top] \\
&= \left(\mathbf{I} - \frac{\eta^2}{2}\boldsymbol{\Lambda}_{\text{pos}}^{\beta_k}\right)\boldsymbol{\Sigma}_{k-1}\left(\mathbf{I} - \frac{\eta^2}{2}\boldsymbol{\Lambda}_{\text{pos}}^{\beta_k}\right)^\top + \left(\eta\mathbf{I} - \frac{\eta^3}{4}\boldsymbol{\Lambda}_{\text{pos}}^{\beta_k}\right)\mathbb{E}[\mathbf{v}_{k-1}\mathbf{v}_{k-1}^\top]\left(\eta\mathbf{I} - \frac{\eta^3}{4}\boldsymbol{\Lambda}_{\text{pos}}^{\beta_k}\right)^\top \\
&= \left(\mathbf{I} - \frac{\eta^2}{2}\boldsymbol{\Lambda}_{\text{pos}}^{\beta_k}\right)\boldsymbol{\Sigma}_{k-1}\left(\mathbf{I} - \frac{\eta^2}{2}\boldsymbol{\Lambda}_{\text{pos}}^{\beta_k}\right)^\top + \eta^2\mathbf{I} - \frac{\eta^4}{4}\boldsymbol{\Lambda}_{\text{pos}}^{\beta_k} - \frac{\eta^4}{4}\boldsymbol{\Lambda}_{\text{pos}}^{\beta_k} + \frac{\eta^6}{16}\left(\boldsymbol{\Lambda}_{\text{pos}}^{\beta_k}\right)^2 \\
&= \left(\mathbf{I} - \frac{\eta^2}{2}\boldsymbol{\Lambda}_{\text{pos}}^{\beta_k}\right)(\boldsymbol{\Sigma}_{k-1} - \boldsymbol{\Sigma}_{\text{pos}}^{\beta_k})\left(\mathbf{I} - \frac{\eta^2}{2}\boldsymbol{\Lambda}_{\text{pos}}^{\beta_k}\right) + \boldsymbol{\Sigma}_{\text{pos}}^{\beta_k} - \frac{\eta^4}{4}\boldsymbol{\Lambda}_{\text{pos}}^{\beta_k} + \frac{\eta^6}{16}\left(\boldsymbol{\Lambda}_{\text{pos}}^{\beta_k}\right)^2
\end{aligned}
$$
(75)

where we leverage the fact that $\boldsymbol{\mu}_{k-1}$ and $\mathbf{v}_{k-1}$ are independent and hence uncorrelated under full momentum refreshment, and that $\mathbf{v}_{k-1} \sim \mathcal{N}(\mathbf{0}, \mathbf{I})$ so $\mathbb{E}[\mathbf{v}_{k-1}\mathbf{v}_{k-1}^\top] = \mathbf{I}$.

For the mean of the updated momentum, we have

$$
\boldsymbol{\mu}_k^{\mathbf{v}} = \mathbb{E}_{q_{\text{fwd}}}\left[\left(\mathbf{I} - \frac{\eta_k^2}{2}\boldsymbol{\Lambda}_{\text{pos}}^{\beta_k}\right)\mathbf{v}_{k-1} - \eta_k\boldsymbol{\Lambda}_{\text{pos}}^{\beta_k}\boldsymbol{\theta}_{k-1} + \eta_k\boldsymbol{\Lambda}_{\text{pos}}^{\beta_k}\boldsymbol{\mu}_{\text{pos}}^{\beta_k}\right] = \eta_k\boldsymbol{\Lambda}_{\text{pos}}^{\beta_k}(\boldsymbol{\mu}_{\text{pos}}^{\beta_k} - \boldsymbol{\mu}_{k-1}) \quad (76)
$$

where we again use $\mathbb{E}_{q_{\text{fwd}}}[\mathbf{v}_{k-1}] = \mathbf{0}$.

For the covariance of the updated momentum, we have

$$
\begin{aligned}
\boldsymbol{\Sigma}_k^{\mathbf{v}} &= \mathbb{E}_{q_{\text{fwd}}}\left[(\hat{\mathbf{v}}_k - \boldsymbol{\mu}_k^{\mathbf{v}})(\hat{\mathbf{v}}_k - \boldsymbol{\mu}_k^{\mathbf{v}})^\top\right] \\
&= \left(\mathbf{I} - \frac{\eta_k^2}{2}\boldsymbol{\Lambda}_{\text{pos}}^{\beta_k}\right)\mathbb{E}_{q_{\text{fwd}}}\left[\mathbf{v}_{k-1}\mathbf{v}_{k-1}^\top\right]\left(\mathbf{I} - \frac{\eta_k^2}{2}\boldsymbol{\Lambda}_{\text{pos}}^{\beta_k}\right) + \eta_k^2\boldsymbol{\Lambda}_{\text{pos}}^{\beta_k}\boldsymbol{\Sigma}_{k-1}\boldsymbol{\Lambda}_{\text{pos}}^{\beta_k} \\
&= \left(\mathbf{I} - \frac{\eta_k^2}{2}\boldsymbol{\Lambda}_{\text{pos}}^{\beta_k}\right)^2 + \eta_k^2\boldsymbol{\Lambda}_{\text{pos}}^{\beta_k}\boldsymbol{\Sigma}_{k-1}\boldsymbol{\Lambda}_{\text{pos}}^{\beta_k}
\end{aligned}
$$
(77)

where we again leverage the independence of $\mathbf{v}_{k-1}$ and $\boldsymbol{\theta}_k$, and $\mathbb{E}[\mathbf{v}_{k-1}\mathbf{v}_{k-1}^\top] = \mathbf{I}$.

### B.4  Noisy Model for Bayesian Linear Regression

In this section, we justify the additive noise model we used in analyzing the stochastic version of DAIS. In particular, the mini-batch gradient has the following form:

$$
\tilde{\nabla}_{\boldsymbol{\theta}} \log f_{\beta_k}(\boldsymbol{\theta}) = -\boldsymbol{\Lambda}_p(\boldsymbol{\theta} - \boldsymbol{\mu}_p) + \frac{\beta_k n}{\sigma^2}\mathbf{x}\left(y - \mathbf{x}^\top\boldsymbol{\theta}\right), \quad (78)
$$

where $(\mathbf{x}, y)$ is one training sample and $n$ is the number of training samples in the dataset. Compared to the full-batch gradient in (61), we have

$$
\tilde{\nabla}_{\boldsymbol{\theta}} \log f_{\beta_k}(\boldsymbol{\theta}) - \nabla_{\boldsymbol{\theta}} \log f_{\beta_k}(\boldsymbol{\theta}) = \frac{\beta_k}{\sigma^2}\left(n\mathbf{x}\mathbf{x}^\top - \mathbf{X}^\top\mathbf{X}\right)\boldsymbol{\theta} + \frac{\beta_k}{\sigma^2}\left(n\mathbf{x}y - \mathbf{X}^\top\mathbf{y}\right) \quad (79)
$$

As long as the problem is not linearly solvable, i.e., $\mathbf{y} = \mathbf{X}\boldsymbol{\mu}^* + \boldsymbol{\varepsilon}$, we have

$$
\tilde{\nabla}_{\boldsymbol{\theta}} \log f_{\beta_k}(\boldsymbol{\theta}) - \nabla_{\boldsymbol{\theta}} \log f_{\beta_k}(\boldsymbol{\theta}) = \frac{\beta_k}{\sigma^2}\left(n\mathbf{x}\mathbf{x}^\top - \mathbf{X}^\top\mathbf{X}\right)(\boldsymbol{\theta} - \boldsymbol{\mu}_*) + \frac{\beta_k}{\sigma^2}\left(n\mathbf{x}\varepsilon - \mathbf{X}^\top\boldsymbol{\varepsilon}\right) \quad (80)
$$

We typically refer to the first term in (80) as multiplicative noise as it depends on $\boldsymbol{\theta} - \boldsymbol{\mu}_*$ and the second term as additive noise.

## C  Implementation Details and Additional Results

### C.1  Implementation Details for Training Experiments

The prior $p(\mathbf{z})$ is a 50-dimensional standard Gaussian distribution. The conditional distributions $p(\mathbf{x}_i|\mathbf{z})$ are independent Bernoulli, with the decoder parameterized by two hidden layers, each with 200 tanh units. The variational posterior $q(\mathbf{z}|\mathbf{x})$ is also a 50-dimensional Gaussian distribution with diagonal covariance, whose mean and variance are both parameterized by two hidden layers with

200 tanh units. For training, we used Adam [Kingma and Ba, 2015] optimizer for 1,500 epochs with initial learning rate $0.001$ and we decay the learning rate by factor $0.8$ every 100 epochs. By default, we use constant step size and partial meomentum refreshment $\gamma = 0.9$ for all iterations. For each setting, we tune step-size $\eta$ by grid search with the search range $\{0.02, 0.04, 0.06, 0.08, 0.10\}$.

For DAIS (adapt), we learn the annealing scheme along with VAE parameters. In particular, the annealing scheme is parameterized as $\beta_k = \sum_{i=1}^k p_i$ with $p = \mathrm{softmax}(z)$ where $z \in \mathbb{R}^K$ is the trainable parameters. Using this parameterization, we guarantee that $\beta_i \geq 0$, $\beta_K = 1$ and $\beta_{i+1} \geq \beta_i$. To avoid collapse, we add a small amount of entropy regularization on $p$ with a coefficient of $0.01$. With a stronger entropy regularizer, the learned annealing scheme would be closer to linear scheme $\beta_k = \frac{k}{K}$. We note that the performance is insensitive to the entropy regularization coefficient.

### C.2 Implementation Details for Evaluation Experiments

For AIS, we run 10 leapfrog steps (followed by MH accept-reject steps) for every intermediate distribution to be consistent with Wu et al. [2016]. By default, we use linear annealing scheme with $\beta_k = \frac{k}{K}$. For HAIS and DAIS, we use constant step-size and partial momentum refreshment with $\gamma = 0.9$ for simulations. Importantly, we only tune the step-size for $K = 10$ and then we employ the optimal scaling scheme we derived in Theorem 1 with $c = \frac{1}{4}$. By grid search, we find $\eta = 0.08$ is good overall for all the models with $K = 10$, so we use the step-size $\eta$ defined as $0.08 \times (K/10)^{-0.25}$ for any run with $K$ intermediate distributions. For AIS, its step-size is adapted throughout the course of training with a target accept rate of $0.65$ in the MH step. We increase the step-size by multiplying $1.02$ when the accept rate is larger then $0.65$, otherwise we multiply it with factor $0.98$. For all curves, we average over 10 runs.

## D   Notes on Memory-Efficient DAIS

---

**Algorithm 2** Reversible DAIS (forward)

---

**Require:** seed $s_0$, initial state $\boldsymbol{\theta}_0 \sim p_0(\boldsymbol{\theta})$, $\mathbf{v}_0 \sim \pi \triangleq \mathcal{N}(\mathbf{0}, \mathbf{M})$
   **for** $k = 1, \ldots, K$ **do**
      $\boldsymbol{\theta}_{k-\frac{1}{2}} \leftarrow \boldsymbol{\theta}_{k-1} + \frac{\eta_k}{2}\mathbf{M}^{-1}\mathbf{v}_{k-1}$
      $\hat{\mathbf{v}}_k \leftarrow \mathbf{v}_{k-1} + \eta_k \nabla \log f_{\beta_k}(\boldsymbol{\theta}_{k-\frac{1}{2}})$
      $\boldsymbol{\theta}_k \leftarrow \boldsymbol{\theta}_{k-\frac{1}{2}} + \frac{\eta_k}{2}\mathbf{M}^{-1}\hat{\mathbf{v}}_k$
      $s_k \leftarrow \mathrm{FORWARD\_SEED}(s_{k-1})$
      $\boldsymbol{\varepsilon}_k \sim \mathcal{N}(\mathbf{0}, \mathbf{M}; s_k)$
      $\mathbf{v}_k \leftarrow \gamma \hat{\mathbf{v}}_k + \sqrt{1-\gamma^2}\boldsymbol{\varepsilon}_k$
   **end for**
   **return** $\boldsymbol{\theta}_K, \mathbf{v}_K, s_K$

---

**Algorithm 3** Reversible DAIS (backward)

---

**Require:** $s_K, \boldsymbol{\theta}_K, \mathbf{v}_K$
   **for** $k = K, \ldots, 1$ **do**
      $\boldsymbol{\varepsilon}_k \sim \mathcal{N}(\mathbf{0}, \mathbf{M}; s_k)$
      $\hat{\mathbf{v}}_k \leftarrow \frac{1}{\gamma_k}\left(\mathbf{v}_k - \sqrt{1-\gamma^2}\boldsymbol{\varepsilon}_k\right)$
      $s_{k-1} \leftarrow \mathrm{BACKWARD\_SEED}(s_k)$
      $\boldsymbol{\theta}_{k-\frac{1}{2}} \leftarrow \boldsymbol{\theta}_k - \frac{\eta_k}{2}\mathbf{M}^{-1}\hat{\mathbf{v}}_k$
      $\mathbf{v}_{k-1} \leftarrow \hat{\mathbf{v}}_k - \eta_k \nabla \log f_{\beta_k}(\boldsymbol{\theta}_{k-\frac{1}{2}})$
      $\boldsymbol{\theta}_{k-1} \leftarrow \boldsymbol{\theta}_{k-\frac{1}{2}} - \frac{\eta_k}{2}\mathbf{M}^{-1}\mathbf{v}_{k-1}$
   **end for**
   **return** $\boldsymbol{\theta}_0, \mathbf{v}_0, s_0$

---

Algorithms 2 and 3 detail the simulation of DAIS dynamics in a reversible manner. The momentum refreshment step requires the reversible computation of a seed in order to retrieve noise samples. Reversibility is sufficient to facilitate reverse-mode differentiation through the chain without explicitly storing the DAIS trajectory. Hence, DAIS can be made memory-efficient while retaining differentiability to any of its parameters.

However, as detailed by Maclaurin et al. [2015], the division by $\gamma_k$ for the computation of $\hat{\mathbf{v}}_k$ in the backward simulation is problematic for finite-precision computation as information is lost with each step. This is combated with Algorithm 3 [Maclaurin et al., 2015] at the minute cost of $\log_2(1/\gamma)$ bits per parameter per step.