# OpenReview forum: "Differentiable Annealed Importance Sampling and the Perils of Gradient Noise"
_NeurIPS.cc/2021/Conference — NeurIPS 2021 Poster_

### Official Review · Reviewer_m6Nw · 2021-07-08

**Rating:** 8
**Confidence:** 2

**Summary:**

The authors develop a variant of annealed importance sampling (AIS) that allows for pathwise derivatives to be computed. This method can be used for (approximate) maximum likelihood learning of model hyperparameters. The authors prove that the proposed algorithm is consistent when applied to Bayesian linear regression (even when transitions are not perfect) and leads to a rate of convergence of $O(1/\sqrt{K})$, where $K$ is the number of transitions performed. The authors show that, although it at first appears the proposed algorithm can be minibatched, the algorithm is inconsistent in the presence of additive noise when applied to Bayesian linear regression.

**Ethics Review Area:**

["I don’t know"]

**Limitations And Societal Impact:**

The authors address limitations of their work well including:
- Limitations in the stochastic setting
- Potential memory limitations in finite precision
- Experiments are not claimed to be state of the art, but illustrate the method and theory well

The authors claim that since the contribution is methodological and theoretical, determining any direct societal impact would be difficult. This seems reasonable for this paper.


**Main Review:**

## Summary of review
The proposed algorithm appears to address a signficant drawback of AIS in a sensible way. The exposition in the paper is generally very good. The analysis of the algorithm in the cases of Bayesian linear regressoin provides some theoretical evidence that the algorithm is practial. The experiments are a useful illustration of the proposed approach, and the authors are very explicit that the experiments are intended as illustrative. Most of my comments, given below, are relatively minor. I am not very familiar with some of the related work, but my impression is that the work contains an interesting contribution, is well-presented, and considers interesting questions relating to the proposed method.

## Notation
- At points the authors refer to $p(\theta, D| \mathcal{M})$ as the target distribution. It seems to me that $p(\theta|D, \mathcal{M})$ is the target distribution (of course these are the same up to a normalizing constant).
- Punctuation is missing from some equations (11, 12).

## Remarks
- I found the justification for the additive noise model slightly unclear. While it seems to have precedent in the literature, it would be nice if the authors could be more explicit about why neglecting the multiplicative term is reasonable. Also is the noise known to satisfy sufficient criteria for the CLT to apply? It seems there is at least some weak dependence between sampled points due unless mini-batches are sampled with replacement. I ask the latter question more out of curiousity, since the precedent for this model in the literature seems quite strong.
- I found the citation to Robbins and Munro in line 244 confusing. If I understand correctly, in your proof it is needed that the step size sequence is not in $\ell_2$, where Robbins and Munro require that it is not in $\ell_1$ but is in $\ell_2$. Please clarify if this is not correct. The placement of this citation led me to think you were claiming that Robbins and Munro require that the sequence of step sizes is in not in $\ell_2$.
- The description of data generating for BLR is confusing. It sounded to me as though $X$ and $Y$ are generated independently, but glancing at the code provided, it seems that $Y$ was generated from the model (i.e. by multiplying $X$ by a random weight vector).

## Questions
- Does minimizing the variational variant of the bound (eqn 14) with respect to variational parameters $\phi$ have an interpretation as appproximate inference? For example, is the difference between the log marginal likelihood and this bound a divergence between $q_\phi$ and the posterior?
- Is there a technical obstacle to extending your proof to $\gamma > 0$, or is the choice $\gamma=0$ made only for simplicity?
- What is meant by "optimal" in theorem 1? Is this matching a known lower bound? Or is it simply indicating that $c=1/4$ leads to the tightest bound for any $c$?
- Could the authors clarify what is meant by equation 42 in the appendix? Is the integral over $R^{d\times d}$ as suggested by $d\Sigma$, or is it over some parameter that parameterizes the matrix $\Sigma$? (This may be standard notation in some areas, if so I would be happy for the authors to point to a reference on the subject).

**Time Spent Reviewing:**

5 hours

---

> ### Author Response · Authors · 2021-08-09
> **Response**
>
> Thank you for your thorough review and many insightful comments. We address your concerns and questions in order below.
>
> - In terms of notation, we will update the paper in the final version. Thank you for bringing this up.
> - Regarding the use of the additive noise model: we show that the noise covariance is lower bounded by that under an additive noise model (which is easier to analyze, see footnote 3). Since our contribution is a negative result (lower bound on the error), it suffices to prove our claim with the additive noise model.
> - We cite Robbins and Munro to convey the high-level intuition. In practice, one can think of $\eta^2$ as the step size in stochastic optimization (wonder if this difference confused you), and then the same argument of Robbins would apply.
> - In terms of the data generation in BLR, it won’t make any difference by generating $X$ and $Y$ independently or generating $Y$ from the model. In the code, we indeed generate the data using the model, but that’s not essential.
> - We believe our DAIS bound does have an interpretation as approximate inference because it is essentially importance sampling. So one can adopt the analysis in [1] to get the variational posterior.
> - Our proof can be extended to the case of $\gamma > 0$, but the proof would be very messy in order to handle the interaction between $v$ and $\theta$. So we focused on $\gamma = 0$ for simplicity.
> - In theorem 1, by optimal we meant the value of $¼$ gives the best rate. First, our analysis only holds for $c \geq ¼$, any value larger than $¼$ leads to a worse rate.
> - For equation 42, one has to choose a path $\Sigma(t)$ for the integral (though the integral is independent of the path because we have an antiderivative), so it’s a one-dimensional integral. Sorry for the confusion, we will make that clear in the next version.
>
> Reference:
> [1] Importance Weighting and Variational Inference.

---

> > ### Comment · Reviewer_m6Nw · 2021-08-16
> > **Reply to author response**
> >
> > Thank you for the answers and clarifications. The questions in my review were well addressed.

---

### Official Review · Reviewer_SRxf · 2021-07-15

**Rating:** 9
**Confidence:** 3

**Summary:**

In this paper, the authors introduce a new *differentiable* annealed importance sampling scheme (DAIS), which provides a lower bound to the log marginal likelihood. Crucially, DAIS *does not* require a Metropolis-Hasting (MH) correction step which is what allows it to be easily differntiated through. This is also combined with variational inference which they title DAVI.

**Limitations And Societal Impact:**

The authors have adequately addressed the limitations and potential negative societal impact of their work.

**Main Review:**

# Strengths

This is a *fantastic* and very well-written paper. Firstly, the significance of this work is huge: AIS is a popular way to estimate the log marginal likelihood of posteriors but they rely on using a sequence of transition kernels, $\mathcal{T}_k$ that has $p_k$ as an invariant distribution. The invariant distribution condition is usually enforced by a MH correction step; it is the MH step that prevents differentiation through AIS. The authors completely side-step this by using Hamiltonian Monte Carlo without a correction step, thus violating the invariant distribution condition. As AIS is a form of IS (which provides an unbiased estimate of the marginal likelihood), it is no surprise that DAIS produces a lower bound to the log marginal likelihood; crucially though, the use of HMC without the MH correction allows for a closed-form and simple expression for the actual estimate of the log marginal likelihood. This allows DAIS to be simply combined with variational inference to produce DAVI.

What I think makes this paper really great is that the authors don't just stop here. The authors investigate the theoretical properties of DAIS on Bayesian linear regression. Amazingly, the authors demonstrate that the log marg likelihood estimate produced by DAIS is consistent as the $K \rightarrow \infty$ for Bayesian linear regression *without the restrictive assumptions used in Neal's AIS paper which relies on the invariance condition of the transition kernel and that each of samples are independent from one another*. Surprisingly, the authors demonstrate that this same result does not apply if stochastic gradients are used (where the stochasticity is from using mini-batches) and their empirical experiments demonstrate that it can provide an arbitrarily bad estimate. This is a huge result and contrasts with the convergence results for stochastic Langevin MCMC and stochastic HMC.

Lastly, the authors applied DAVI on learning a VAE on MNIST where they demonstrated that it is competitive with a vanilla VAE and IWAE. The authors also demonstrated that DAIS can produce competitive log marginal likelihood estimates.

# Weaknesses

In my opinion, there are no glaring weaknesses in this work. There are a couple of things that can be changed to make the paper better but I will list them in the section below.

# Questions/Comments
1) As the variational approximation used in DAVI is more expressive than a simple Gaussian, it would be cool to see how well DAVI does at estimating complex distributions.
2) For the VAE experiments, no mini-batches were used, correct? If not, it would be interesting to see how much worse the learned VAE is when using mini-batches.
3) Errors bars are missing for figures 1 and 21
4) I think some tweaks can be done to the derivations in the appendix to make it more readable. For example, there are many inequalities that would make the derivation easier to follow if they were either said outright or if a reference was provided for instance $\Vert AB\Vert_2 \leq \Vert A \Vert_2 \Vert B \Vert_2$. I also thinking color coding would help a lot as well.

# Conclusion
 As I stated earlier, I think this is a great paper and a valuable contribution to the research community.


**Time Spent Reviewing:**

7 hours

---

> ### Author Response · Authors · 2021-08-09
> **Response**
>
> Thank you for your constructive feedback and kind words about our work. We address your concerns and questions below.
>
> - We will include a visualization that showcases the ability of DAVI on fitting a complex posterior.
> - For VAE experiments, we use a mini-batch of size 128. But the gradient is exact because the partition function (or marginal likelihood) estimation problem is independent for each data point, and each of these estimation problems uses exact gradients. The problem with gradient noise arises in the setting of marginal likelihood estimation when the log-likelihood gradient itself is estimated stochastically.
> - In our experiments, the variance of different runs is very small, so we didn’t include error bars in the paper. The solid lines in figure 1 are the exact computation of our DAIS bound, so there is no randomness.
> - Thank you for the feedback on the readability of our proofs! We will improve that in the next version.

---

> > ### Comment · Reviewer_SRxf · 2021-08-30
> > **Response to authors**
> >
> > Thank you for responding and thank you for such a well-written paper!

---

### Official Review · Reviewer_8rku · 2021-07-16

**Rating:** 7
**Confidence:** 4

**Summary:**

This paper develops and studies a variant of annealed importance sampling with Hamiltonian Monte Carlo transitions that does not require running a Metropolis-Hastings accept/reject step after each HMC proposal.

The proposed algorithm, like AIS, produces unbiased estimates of a model's marginal likelihood. As such, it can be used to define a Monte Carlo variational objective $L_{DAVI}$, as the expected log marginal likelihood estimate. Without the discrete MH accept/reject step, the objective's gradient can be estimated unbiasedly using the reparameterization trick. In an experiment, the authors exploit this to train VAEs using the $L_{DAVI}$ objective, achieving results competitive with other Monte Carlo variational objectives, namely IWAE.

A downside of the proposed algorithm is that without the MH corrections, the consistency of the AIS algorithm (in the long-chain, fine-annealing-schedule limit) is no longer guaranteed. The paper nonetheless proves a consistency result for the modified algorithm when applied to a particular Bayesian linear regression model, and demonstrates empirically that on the VAE task, performance does improve with longer chains. The paper also shows that the algorithm is *not* consistent for the Bayesian linear regression model if the HMC transitions use noisy gradients. These theoretical results are tested and illustrated in an experiment on the Bayesian linear regression model.

**Limitations And Societal Impact:**

Yes; the paper does not overstate its contributions and I think its focus is sufficiently theoretical that if negative societal impact results, it will be very indirectly.

**Main Review:**

This is a nice piece of work: it introduces a clever modification of AIS that makes it suitable for gradient-based variational optimization, presents some evidence that the method works, and makes a couple interesting theoretical observations.

First, what I like about the paper: DAIS strikes me as interesting in part because, at first glance, removing the Metropolis-Hastings correction in AIS should be dangerous. For one, in many MCMC transition kernels, the MH correction plays an essential role, and without it, the AIS extended proposal distribution could be quite far from the target, yielding high-variance marginal likelihood estimates. Furthermore, the stationarity of the MCMC kernels in AIS leads to a natural choice of extended target distribution, and a simple, easy-to-work-with expression for the marginal likelihood estimate—ditching the MH correction risks losing that. But the paper's clever choice to use Hamiltonian Monte Carlo transitions sidesteps both concerns: well-tuned HMC proposals are almost always accepted anyway, so removing the MH step should not really affect the inference distribution; and the authors show that even without the MH step it is possible to choose a simple extended target $q_{bwd}$, for which the resulting importance weight is a "surprisingly simple expression" (L132). (I was surprised!)

It is also nice to see an analysis of the algorithm, even if only for a specific model (the Bayesian linear regression model)—consistency in the long-chain limit is another thing you might expect to lose by dropping the MH steps, and it's interesting to have an example where it still holds.

A potential weakness of the paper is that its aims are somewhat unclear, as partly evidenced by the title: is the goal to present a new method for variational inference, illuminate "perils of gradient noise," or something else altogether? Because of this, it's unclear by what standards the submission should be judged. For instance:

* I'm not sure what exactly the takeaway message is regarding gradient noise. Even outside an AIS setting, it is known that naively using stochastic gradients in Hamiltonian Monte Carlo can be arbitrarily bad (https://arxiv.org/abs/1402.4102). Furthermore, using stochastic gradients changes the distribution $q_{fwd}$; is there a corresponding change to $q_{bwd}$ that preserves their density ratio $L_{DAIS}$? If not, I'd expect $L_{DAIS}$ to be biased, for any chain length. (If there *is* an importance sampling justification for $L_{DAIS}$ in the presence of gradient noise, it could be useful to present it.)

* If the goal of the paper were primarily to introduce a new method, it would be nice to see more empirical validation or at least discussion of the method's value. What does DAVI add over IWAE? Intuitively, AIS encodes a much more sophisticated inference process than IWAE does (MCMC can solve many inference problems that would be difficult to solve with vanilla importance sampling); are there interesting model families that require this flexibility in the variational posterior? Are there other interesting applications of a differentiable AIS algorithm?

Overall, however, I am inclined to accept the paper because it makes an interesting and technically sound contribution that I can imagine others in the NeurIPS community appreciating and building on.

**Time Spent Reviewing:**

5 hours

---

> ### Author Response · Authors · 2021-08-09
> **Response**
>
> Thank you for your thoughtful review and many amazing questions. We address your questions/comments in order below.
>
> - In terms of the focus of our paper, we make three separate contributions: a new algorithm, a consistency analysis, and the negative result in the stochastic setting. We believe all three parts are interesting, and directly reinforce each other, which is why we submitted all of these contributions as one paper.
> - Our conclusion that DAIS is incompatible with stochastic gradients is surprising even in light of the cited paper on SGHMC. That paper shows that SGHMC with mini-batch gradients is inconsistent in the sense that it has the wrong stationary distribution, but the bias in the stationary distribution vanishes in the limit of small step sizes. What’s different about our DAIS analysis is that the error in the log-ML estimates can’t be eliminated even by taking smaller steps and/or more steps. The reason for this difference is that the SGHMC analysis focuses on last-iteration convergence (and indeed, our Bayesian linear regression analysis also allows for last-iteration convergence in the stochastic setting with decaying step sizes, i.e., both $\mu_K \rightarrow \mu_{\text{pos}}$ and $\Sigma_K \rightarrow \Sigma_\text{pos}$) while we analyze the error in the log-ML estimate, which depends on all states in the chain. So these two negative results are fundamentally different.
> - In terms of importance sampling justification, conditioning on the same mini-batch of data (i.e., the reverse transition $\tilde{\mathcal{T}}_k$ is done using the same batch as $\mathcal{T}_k)$, we can retain the formalism of doing importance sampling on an extended space. This suggests that our DAIS bound (equation 11) is still valid even in the stochastic setting. Intuitively, the reason why our DAIS bound is not consistent in the presence of gradient noise is that the mini-batch sampling of data results in variations in the generated trajectories. Importantly, our DAIS bound depends on the whole trajectory, so using decaying step sizes (to eliminate the last-iteration variation) is not enough (this is related to your last question). We will clarify that in the next version, thank you for bringing this up and this is a fantastic question!
> - For the value of our DAVI over IWAE, we first note that one can perform importance sampling using multiple DAVI chains, just as with AIS (see equation 22). Therefore, DAIS can be seen as an additional degree of freedom which can be combined with the IWAE. More importantly, it’s well known that naive importance sampling can have exponential sample complexity in some problem parameter (e.g. the dimension), and for some problems, AIS can overcome this to give accurate results efficiently (see, for instance, the analysis in [1].). We expect the advantages of AIS to carry over to DAVI as well. In practice, we found that DAVI performs better than IWAE with a large compute budget (see our experiments in VAE), even if IWAE can be a good option when compute is very limited. For large VAE models, the posterior can be very complicated, we believe our DAVI can be adopted to improve the performance.
> - In terms of application, DAIS can save us from tuning annealing, sampling, or many other hyperparameters for inference tasks. In addition, it could be potentially used to optimize the prior in Bayesian neural networks, PAC-Bayes bounds, and Bayesian meta-learning.
>
> Reference:
> [1] Annealed Importance Sampling.

---

### Decision · Program_Chairs · 2021-09-27

**Decision:**

Accept (Poster)

**Comment:**

This paper suggests a method for using annealed importance sampling (AIS) together with Hamiltonian Monte Carlo (HMC). The central idea is that if the accept/reject step is dropped, once can still define a ratio of augmented distributions that is tractable to estimate. The advantage of dropping the accept/reject step is that the estimator is now differentiable. A variational objective is defined in terms of this estimator, and reparameterization is used to optimize the objective. One might worry that dropping the accept/reject step could have serious consequences by changing the properties of the estimator. This is addressed, at least for the case of a simple Bayesian linear regression model, by an analysis that shows a convergence rate of 1/\sqrt{K} can be obtained, where K is the number of annealing distributions. Another analysis is given where gradients are corrupted with additive noise. Here, a more negative result is given that even in the limit that K goes to infinity, the variational bound never becomes tight. This suggests that a significant price is paid for computing gradients on subsets of data. Strictly speaking, this negative result also applies only to the Bayesian linear regression setting, however this shows that any stronger guarantee would have to exclude that-typically considered "easiest"-setting.

Generally speaking, all reviewers felt the algorithm was novel, correct, and potentially significant. There were some fairly minor technical concerns about the experiments, and some questions about the intended "scope" of the contribution (e.g. regarding the title). The authors appear to have good answers for these questions that could easily be integrated into the paper for a final submission.